



# Measurement of iodine species and sulfuric acid using bromide chemical ionization mass spectrometers

Mingyi Wang[1,2*], Xu-Cheng He[3*#], Henning Finkenzeller[4], Siddharth Iyer[3], Dexian Chen[1,5], Jiali Shen[3], Mario Simon[6], Victoria Hofbauer[1,2], Jasper Kirkby[6,7], Joachim Curtius[6], Norbert Maier[8], Theo Kurtén[3,8], Douglas R. Worsnop[3,9], Markku Kulmala[3,10,11,12], Matti Rissanen[3,13], Rainer Volkamer[4], Yee Jun Tham[3#], Neil M. Donahue[1,2,5,14], and Mikko Sipilä[3]

[1]Center for Atmospheric Particle Studies, Carnegie Mellon University, Pittsburgh, PA, 15213, USA
[2]Department of Chemistry, Carnegie Mellon University, Pittsburgh, PA, 15213, USA
[3]Institute for Atmospheric and Earth System Research (INAR), University of Helsinki, 00014 Helsinki, Finland
[4]Department of Chemistry & CIRES, University of Colorado Boulder, Boulder, CO 80309, USA
[5]Department of Chemical Engineering, Carnegie Mellon University, Pittsburgh, PA, 15213, USA
[6]Institute for Atmospheric and Environmental Sciences, Goethe University Frankfurt, 60438 Frankfurt am Main, Germany
[7]CERN, the European Organization for Nuclear Research, CH-1211 Geneve 23, Switzerland
[8]Department of Chemistry, University of Helsinki, 00014 Helsinki, Finland
[9]Aerodyne Research, Inc., Billerica, MA, 01821, USA
[10]Helsinki Institute of Physics, P.O. Box 64 (Gustaf Hallstromin katu 2), FI-00014 University of Helsinki, Finland
[11]Joint International Research Laboratory of Atmospheric and Earth System Sciences, Nanjing University, Nanjing, China
[12]Aerosol and Haze Laboratory, Beijing Advanced Innovation Center for Soft Matter Science and Engineering, Beijing University of Chemical Technology, Beijing, China
[13]Aerosol Physics Laboratory, Physics Unit, Faculty of Engineering and Natural Sciences, Tampere University, Tampere, Finland
[14]Department of Engineering and Public Policy, Carnegie Mellon University, Pittsburgh, PA, 15213, USA
*These authors contributed equally to this work

#Correspondence to:
Xu-Cheng He (xucheng.he@helsinki.fi) and Yee Jun Tham (yee.tham@helsinki.fi).

**Abstract.** Iodine species are important in the marine atmosphere for oxidation and new-particle formation. Understanding iodine chemistry and iodine new-particle formation requires high time resolution, high sensitivity, and simultaneous measurements of many iodine species. Here, we describe the application of bromide chemical ionization mass spectrometers (Br-CIMS) to this task. During iodine new-particle formation experiments in the Cosmics Leaving OUtdoor Droplets (CLOUD) chamber, we have measured gas-phase iodine species and sulfuric acid using two Br-CIMS, one coupled to a Multi-scheme chemical IONization inlet (Br-MION-CIMS) and the other to a Filter Inlet for Gasses and AEROsols inlet (Br-FIGAERO-CIMS). From offline calibrations and inter-comparisons with other instruments attached to the CLOUD chamber, we have quantified the sensitivities of the Br-MION-CIMS to HOI, $I_2$, and $H_2SO_4$ and obtain detection limits of $5.8\times10^6$, $6.3\times10^5$, and $2.0\times10^5$ molec cm$^{-3}$, respectively, for a 2-min integration time. From binding energy calculations, we estimate the detection limit for $HIO_3$ to be $1.2\times10^5$ molec cm$^{-3}$, based on an assumption of maximum sensitivity. Detection limits in the Br-FIGAERO-CIMS are around one order of magnitude higher than those in the Br-MION-CIMS; for example, the detection limits for HOI and $HIO_3$ are $3.3\times10^7$ and $5.1\times10^6$ molec cm$^{-3}$, respectively. Our comparisons of the performance of the MION inlet and the FIGAERO inlet

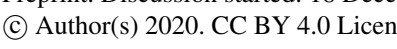



show that bromide chemical ionization mass spectrometers using either atmospheric pressure or reduced pressure
interfaces are well-matched to measuring iodine species and sulfuric acid in marine environments.

**1 Introduction**
Reactive iodine species are released into the atmosphere mainly by biological processes in marine environments (i.e.
from macro- and micro-algae) (McFiggans et al., 2004), $O_3$ deposition on the sea surface (Carpenter et al., 2013), as
well as from the sea ice (Spolaor et al., 2013) and snowpack in the polar region (Raso et al., 2017). Once emitted,
iodine species can modify atmospheric oxidative capacity via a chain of catalytic reactions with $O_3$ that form iodine
oxides, leading to about 20–28 % of $O_3$ loss in the marine boundary layer (Prados-Roman et al., 2015; Sherwen et al.,
2016). Through convection, reactive iodine species can be transported from the lower troposphere to the upper
troposphere-lower stratosphere, causing one third of the iodine-induced ozone loss in the upper troposphere-lower
stratosphere (Koenig et al., 2020). Another important effect of iodine species is their contribution to atmospheric new-
particle formation. O'Dowd et al. (O'Dowd et al., 2002) showed that particles are produced from condensable iodine-
containing vapours at a costal location (Mace Head in Ireland). Recent studies have demonstrated that iodic acid ($HIO_3$)
dominates the charged iodine cluster formation, and drives the bursts of freshly-formed particles in coastal regions
(He et al., 2020; Sipilä et al., 2016). This process thereby may enhance cloud condensation nuclei formation, affecting
climate both directly and indirectly (Saiz-Lopez et al., 2012; Simpson et al., 2015).

Understanding iodine chemistry and iodine driven new-particle formation requires high time resolution, high
sensitivity, and simultaneous measurements of iodine species. However, this has been a long-standing challenge due
to their low abundance and short atmospheric lifetimes. Previous studies have achieved detection of relatively more
abundant molecular iodine ($I_2$), iodine monoxide (IO), and iodine dioxide (OIO) via optical spectroscopy, such as
differential optical absorption (Leigh et al., 2010), cavity ring-down (Bitter et al., 2005), cavity enhanced absorption
(Vaughan et al., 2008), laser-induced fluorescence (Dillon et al., 2006), and resonance fluorescence (Gómez Martín
et al., 2011). The spectroscopic techniques are invaluable; however, their very specificity limits them to the detection
of a few iodine compounds, and they are less sensitive to other iodine species that have congested or broad absorption
cross sections such as hypoiodous acid (HOI) and iodic acid ($HIO_3$).

Another commonly used technique is mass spectrometry; it has a fast response time and a low detection limit, but
extra calibration efforts are needed for the quantification of the detection sensitivity. For example, photoionization
(Gómez Martín et al., 2013) and chemical ionization mass spectrometry (CIMS) have been employed to detect a suite
of halogen species. Reagent ions used with CIMS include: $SF_5^-$ for HCl and $ClONO_2$ (Marcy et al., 2004); iodide ($I^-$)
for atmospheric chlorine and bromine species such as $ClNO_2$, $Cl_2$, ClO, BrO, and BrCl (Kercher et al., 2009; Lee et
al., 2018; Tham et al., 2016); superoxide ($O_2^-$) for molecular iodine ($I_2$) (Finley and Saltzman, 2008); and both nitrate
($NO_3^-$) (Sipilä et al., 2016) and protonated water ($H_3O^+$) (Pfeifer et al., 2020) for $HIO_3$. The nitrate-CIMS and $H_3O^+$-
CIMS suffer from the limited analyte affinity to the reagent ions. The iodide-CIMS can effectively measure chlorine
and bromine species, but it is not suitable to detect iodine species due to the ambiguity in peak identification.



Bromide ion (Br⁻) exhibits an affinity to a wide spectrum of iodine containing species. Br-CIMS has been routinely
used to measure chlorine species (Lawler et al., 2011), $HO_2$ radicals (Sanchez et al., 2016), organic vapors and sulfuric
acid (Rissanen et al., 2019), and nitric acid (Wang et al., 2020). Like chlorine species, iodine species are known to
cluster with bromide ions via halogen (or hydrogen) bonds; as such, here we explore using the Br-CIMS to measure
gas-phase iodine species and sulfuric acid simultaneously at concentrations relavant to the marine boundary layer. In
this study, we demonstrate the detection of various gas-phase inorganic iodine species with the Br-CIMS and explore
the effect of relative humidity (RH) on that detection. We then quantify the sensitivies of several gas-phase halogen
species via inter-method calibration, offline calibration, and quantum chemical calculations. Finally, we compare the
performance of Br-MION-CIMS and Br-FIGAERO-CIMS and show that both of them are well-suited for iodine
species measurement in the atmosphere.

**2 Methodology**
**2.1 The CLOUD facility**
We conducted measurements and instrument inter-comparison at the CERN CLOUD facility, a 26.1 m³
electropolished stainless-steel chamber that enables new-particle formation experiments simulating the typical range
of tropospheric conditions with scrupulous cleanliness and minimal contamination (Duplissy et al., 2016; Kirkby et
al., 2011). The CLOUD chamber is mounted in a thermal housing, capable of keeping temperature constant in a range
of -65 °C and +100 °C with ±0.1 °C precision (Dias et al., 2017) and relative humidity commonly between < 0.5%
and 80%. Photochemical processes are driven by different light sources, including four 200 W Hamamatsu Hg-Xe
lamps with significant spectral irradiance between 250 and 450 nm, and an array of 48 green light LEDs at 528 nm
with adjustable optical power up to 153 W. Ion-induced nucleation under different ionization levels is simulated with
a combination of electric fields (electrodes at ±30 kV at top and bottom of the chamber) which can be turned on to
rapidly scavenge smaller ions, and a high-flux beam of 3.6 GeV pions ($\pi^+$) which enhances ion production when turned
on. Mixing is accelerated with magnetically coupled fans mounted at the top and bottom of the chamber. The
characteristic gas mixing time in the chamber during experiments is a few minutes. The loss rate of condensable vapors
onto the chamber wall is comparable to the condensation sink in pristine boundary layer environments (e.g. $2.2 \times 10^{-3}$
$s^{-1}$ for $H_2SO_4$ at 5 °C). To avoid a memory effect between different experiments, the chamber is periodically cleaned
by rinsing the walls with ultra-pure water and heating to 100 °C for at least 24 hours, ensuring extremely low
contaminant levels of sulfuric acid ($< 5 \times 10^4$ cm$^{-3}$) and total organics (< 150 pptv) (Kirkby et al., 2016; Schnitzhofer
et al., 2014). The CLOUD gas system is also built to the highest technical standards of cleanliness and performance.
The dry air supply for the chamber is provided by cryogenic oxygen (Messer, 99.999 %) and cryogenic nitrogen
(Messer, 99.999 %) mixed at the atmospheric ratio of 79:21. Ultrapure water vapor, ozone and other trace gases can
be precisely added to attain mixing ratios at different levels.

**2.2 Br-MION-CIMS**



We measured gas-phase iodine species with a bromide chemical ionization atmospheric pressure interface time-of-
flight mass spectrometer (Junninen et al., 2010) coupled with a Multi-scheme chemical IONization inlet (Br-MION-
CIMS) (Rissanen et al., 2019). The Br-MION inlet consists of an electrically grounded 24 mm inner diameter stainless
steel flow tube, attached to an ion source. For the CLOUD measurements, the length of the sampling inlet was ~1.5
m and was designed to be in a laminar flow with a fixed total flow rate of 20 standard liters per minute (slpm). An ion
filter, operated with positive and negative voltage, was placed at the front of the inlet to filter out any ions in the
sample air prior to ion-molecule reaction chamber in the inlet. The reagent ions, bromide ($Br^-$) and the bromide-water
cluster ($H_2O \cdot Br^-$), were produced by feeding 25 standard milliliters per minute (mlpm) of nitrogen ($N_2$) flow through
a saturator containing dibromomethane ($CH_2Br_2$; > 99.0 %, Tokyo Chemical Industry) into the ion source, where the
reagent was ionized by soft X-ray radiation. The resulting ions were then accelerated by a 2500 V ion accelerator array
and focused by a 250 V ion deflector into the laminar sampling flow of the inlet via a 5 mm orifice. A small counter
flow (~40 mlpm) was applied through the orifice to prevent any mixing of the electrically neutral reagent vapor with
the sampling flow. The details of the inlet design, setup, and operation are described in Rissanen et al., 2019.
**2.3 Br-FIGAERO-CIMS**
We also measured both the gas- and particle-phase compositions via thermal desorption using a bromide chemical
ionization time-of-flight mass spectrometer equipped with a Filter Inlet for Gases and AEROsols (Br-FIGAERO-
CIMS) (Lopez-Hilfiker et al., 2014). FIGAERO is a manifold inlet for a CIMS with two operating modes. In the
sampling mode, gases are directly sampled into a 150 mbar ion-molecule reactor, using coaxial core sampling to
minimize their wall losses in the sampling line. The total flow is maintained at 18.0 slpm and the core flow at 4.5 slpm;
the CIMS samples at the center of the core flow with a flow rate at ~1.6 slpm. Concurrently, particles are collected on
a PTFE filter via a separate dedicated port with a flow rate of 6 slpm. In the desorption mode, the filter is automatically
moved into a pure $N_2$ gas stream flowing into the ion molecule reactor, while the $N_2$ is progressively heated upstream
of the filter to evaporate the particles via temperature programmed desorption. Analytes are then chemically ionized
by $Br^-$ and extracted into a mass spectrometer.
**2.4 CE-DOAS**
For the quantitative measurement of gas-phase molecular iodine ($I_2$), we deployed a Cavity Enhanced Differential
Optical Absorption Spectroscopy instrument (CE-DOAS) (Meinen et al., 2010). CE-DOAS determines concentrations
of trace gases from the strength of differential spectral features in a reference spectrum. The accuracy of the method
is ultimately determined by the uncertainty of the respective absorption cross sections. It is thus an absolute method
and does not depend on an instrument specific detection efficiency. To maximize the measurement sensitivity towards
$I_2$, we used a setup optimized for the green wavelength range (508-554 nm), where $I_2$ exhibits strong differential
absorption features. The measurement light is provided by a green light emitting diode (LED Engin). Spectral
dispersion is established with a Czerny-Turner grating spectrometer (Princeton Instruments Acton 150), resulting in
an optical resolution of 0.73 nm full width at half maximum at 546 nm. Intensities are monitored with a CCD detector
(charge-coupled device, Princeton Instruments PIXIS400B) cooled to -70 °C. Highly reflective mirrors (Advanced





155 Thin Films) enhance the 1 m mirror separation to an effective optical path length of 15-23 km. The effective spectral

156 mirror reflectivity was established by comparing light intensity spectra in the presence of $N_2$ and He (Washenfelder

157 et al., 2008). The abundance of trace gases is then determined by comparing spectra of chamber air relative to reference

158 spectra recorded with ultrapure $N_2$ without $I_2$. Chamber air is drawn into the cavity with a constant flow rate of 1 slpm.

159 Variations of the sampling flow did not result in changes in measured $I_2$ concentrations, indicating that photolysis

160 from the measurement light within the instrument was negligible. The following absorbers were included in the fit: $I_2$

161 (Spietz et al., 2006), $NO_2$ (Vandaele et al., 1998), $H_2O$ (Rothman et al., 2010), $O_2$-$O_2$ collision-induced absorption

162 (Thalman and Volkamer, 2013), and a polynomial of sixth order. The setup allowed a 1-minute detection limit of 25

163 pptv, or 8 pptv for integration times of 10 minutes, respectively. Periodic automated recordings of $N_2$ reference spectra

164 were recorded to ensure baseline stability. The optical path length at the time of measurement was continuously

165 confirmed for consistency by the measurement of the $O_2$-$O_2$ collision-induced absorption and $H_2O$ column in the same

166 analysis window. The overall systematic accuracy for the $I_2$ time series is estimated to be 20 %, never better than the

167 detection limit, resulting from the uncertainty in cross sections and the stability of the baseline.

168

169 **2.5 Quantum chemical calculations**

170 We used quantum chemical calculations to estimate the cluster formation enthalpy of halogen containing species and

171 bromide ions. The initial conformer sampling was performed using the Spartan '14 program. The cluster geometry

172 was then optimized using density function theory methods at the ωB97X-D/aug-cc-pVTZ-PP level of theory (Chai

173 and Head-Gordon, 2008; Kendall et al., 1992). Iodine pseudopotential definitions were taken from the EMSL basis

174 set library (Feller, 1996). Calculations were carried out using the Gaussian 09 program (Frisch et al., 2010). An

175 additional coupled-cluster single-point energy correction was carried out on the lowest energy geometry to calculate

176 the final cluster formation enthalpy. The coupled-cluster calculation was performed at the DLPNO-CCSD(T)/def2-

177 QZVPP level using the ORCA program ver. 4.0.0.2 (Neese, 2012; Riplinger and Neese, 2013). In Table 1 we present

178 calculated cluster formation enthalpies based on the optimized geometries.

179

180 **3 Results and Discussion**

181 **3.1 Detection of gas-phase inorganic species by Br-MION-CIMS**

182 We show in Fig. 1 the selected inorganic species observed with the Br-MION-CIMS during an iodine new-particle

183 formation experiment in the CLOUD chamber. The peak identities are indicated in the labels. Observed species

184 include $I_2$ and its various oxidation products. There are also a few other halogen-containing inorganic species such as

185 $Cl_2$, ICl and IBr, likely coming from the impurities in the iodine source. Non-halogen inorganic species such as $H_2SO_4$

186 can also react with bromide ion and are detected. Due to the large negative mass defect of the bromine and iodine

187 atoms, and the high resolution (~10000 Th Th$^{-1}$) of the mass spectrometer, the peaks can be unambiguously

188 distinguished and identified in the mass spectrum. As shown in the lower panel of Fig. 1, most of the iodine-containing

189 species appear as a single peak in the unit mass range, except for $HIO_2 \cdot ^{79}Br^-$ (m/z = 238.82), which overlaps with the

190 reagent ion cluster $(^{79}Br_2^{81}Br)^-$ (m/z = 238.75).

191





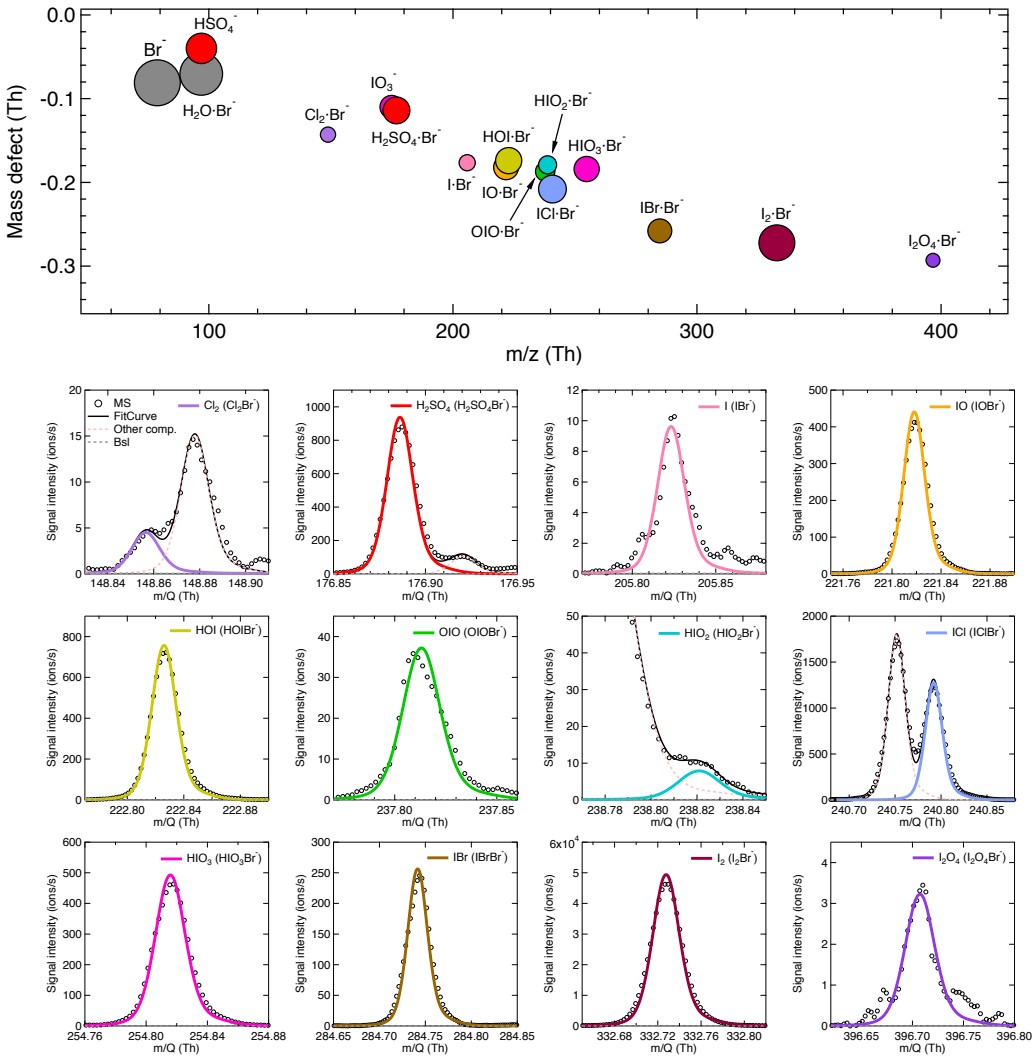

**Figure 1**. Gas-phase inorganic species measured with the Br-MION-CIMS. Mass defect (difference of exact mass to integer mass) versus m/z of gas-phase halogen species and sulfuric acid during an iodine new-particle formation experiment at 69 % relative humidity and -10 °C. Ions shown here are either clustered with or formed via proton transfer to a bromide ion. The area of the markers is proportional to the logarithm of the signal (counts per second). Shown in the lower panel are the high-resolution single peak fits for species in the mass defect plot in the upper panel.

The iodine new-particle formation experiments were conducted under experimental conditions typically found in the high-latitude marine boundary layer, with a temperature of -10 °C and a relative humidity of 69 %. As illustrated in Fig. 2, a typical experiment started with illumination of the chamber at constant $I_2$ (~60 pptv) using the green light to photolytically produce I atoms. The subsequent reactions of I and ~ 40 ppbv $O_3$ led to the formation of various oxidized iodine species within a few minutes. The most prominent species we observed from these experiments were IO, HOI and $HIO_3$, with lower but significant levels of OIO, $HIO_2$, and $I_2O_4$. Among these iodine oxides, IO rose the most

rapidly; this is consistent with the first-generation production of IO from the I + $O_3$ reaction. After a few steps of
radical reactions, OIO, $HIO_2$ and $HIO_3$ reached steady state almost simultaneously. The only observed iodine oxide
dimer was $I_2O_4$ in this event, while $I_2O_2$, $I_2O_3$ and $I_2O_5$ were below the detection limit of both mass spectrometers. A
noticeable dip in the $HIO_3$ traces a few minutes after the onset of the reactions is likely due to the participation of $HIO_3$
in new-particle formation, resulting in an extra loss term and a lower steady-state concentration. When we turned off
the green light, the production of I radicals stopped and iodine species decayed.

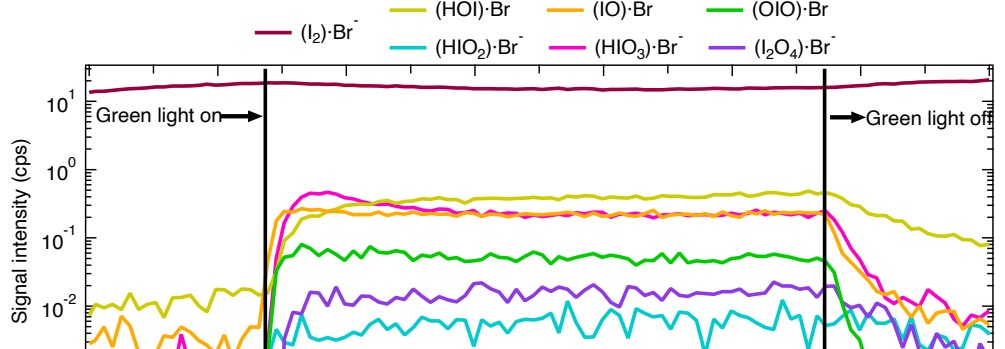


**Figure 2**. Evolution of selected iodine species during a typical run. The experiment was performed at 60 pptv $I_2$, 40 ppbv $O_3$, 69 %
relative humidity and -10 °C. The oxidized iodine species start to appear soon after switching on the green light at 08:11, 05 October
2018. The I atom production was halted at 10:21, 05 October 2018 by switching off the green light, and the concentration of
oxidized iodine species decayed away afterwards. All species are color-coded in the same way as in Fig. 1.

**3.2 Relative humidity dependence**
Water molecules can cluster with $I^-$ to form $H_2O \cdot I^-$ in the iodide CIMS. This enhances the instrument sensitivities for
small molecules (i.e. inorganics) and reduces them for large molecules (i.e. organics) (Lee et al., 2014). To investigate
the role of water concentration in the sensitivity of the Br-MION-CIMS, we varied the relative humidity (RH) from
40 % to 80 % at a constant temperature of -10 °C. We show in Fig. 3 the correlation of $I_2$ time series from the Br-
MION-CIMS and the CE-DOAS throughout the experiment. During the RH transition, the ratio of the two reagent
ions, $Br^-$ and $H_2O \cdot Br^-$, changed in the Br-MION-CIMS. As shown in Fig. 3 (a) and (b), using either reagent ion alone
for $I_2$ normalization results in discrepancies in recovered $I_2$ concentrations at different RH. However, if we use the
sum of these two reagent ions ($Br^-$ + $H_2O \cdot Br^-$) for normalization, the humidity effect vanishes, as shown in the Fig. 3
(c). This suggests that the quantitative detection of $I_2$ molecules is robust and independent of RH, as long as a proper
normalization method is used for the Br-MION-CIMS. Furthermore, we have also carried out the HOI calibration and
used the same normalization method as described in section 3.3.4. During the calibration, we varied the water content
in the calibrator to vary OH concentrations. A good correlation between the modeled HOI concentrations and the
measured HOI signals also indicates that the different $H_2O$ concentrations in the system do not affect the HOI detection.
This assertion may also be applicable to other molecules, but further confirmation is needed.






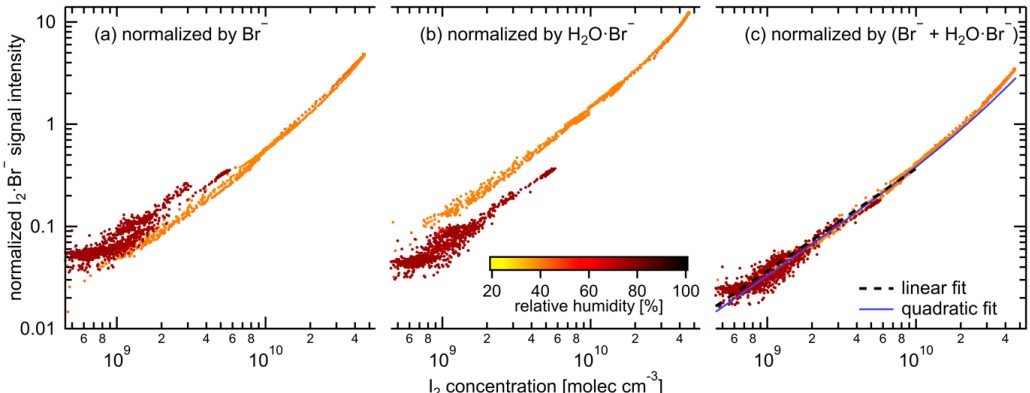

**Figure 3.** Signal normalization methods for the Br-MION-CIMS. The charger ions in the ion source of Br-MION-CIMS are Br⁻
and $H_2O\cdot Br^-$ (both $^{79}Br$ and $^{81}Br$). Their abundance depends both on the instrument tuning and the absolute humidity of the sampled
flow. The normalization of the $I_2\cdot Br^-$ signal by only Br⁻ (a) or $H_2O\cdot Br^-$ (b) does not compensate for the humidity effect. Using the
sum of Br⁻ and $H_2O\cdot Br^-$ (c) for normalization yields a tight correlation to the true $I_2$ as measured by CE-DOAS, independent of the
humidity. The black dashed line and blue solid curve indicate the fitted linear and quadratic calibration curves, respectively.

**3.3 Quantification of gas-phase inorganic species**
**3.3.1 $I_2$ calibration using the CE-DOAS**
As shown in Fig. 3, we use the accurate $I_2$ time series measured with the CE-DOAS to calibrate normalized $I_2$ signals
in the Br-MION-CIMS. The $I_2$ concentrations used for the calibration span approximately 2 orders of magnitude,
reaching up to $4.6 \times 10^{10}$ molec cm⁻³. A linear fit, limited to $I_2$ concentrations smaller than $10^{10}$ molec cm⁻³, establishes
the calibration factor $[I_2] = 2.7 \times 10^{10}$ molec cm⁻³ $\times I_2 \cdot {}^{79}Br^-/({}^{79}Br^- + H_2O\cdot {}^{79}Br^-)$. We also use a quadratic fit to establish
the calibration for the entire range of concentrations encountered during whole campaign (solid line in Fig. 3 (c)); two
curves agree well. The CE-DOAS $I_2$ detection limit is $6.3 \times 10^8$ molec cm⁻³ (25 pptv) for a 1 min integration time, and
the total systematic uncertainty is estimated to be 20%. Deviations between both time series are generally smaller than
10% (25 and 75 percentile 0.88 and 1.03, respectively). These small differences are consistent with incomplete
homogeneity of iodine concentrations in the chamber and the different sampling positions of CE-DOAS and Br-
MION-CIMS.

**3.3.2 $I_2$ calibration using a permeation tube**
We used an iodine permeation tube (VICI Metronic) as a source for offline calibration. The permeation tube was
encased within an electronically controlled heating mantle ($80 \sim 140 (\pm 2)$ °C) to allow for adjustable yet steady iodine
permeation rates. The iodine permeation device was run continuously for at least 72 hours before any calibration
experiments to ensure that a complete equilibrium was reached in the system. We then confirmed the robustness of
the permeation device by the constant $I_2$ signal measured with Br-MION-CIMS for over 24 hours.

To determine the permeation rate of $I_2$, we trapped iodine in n-hexane at cryogenic temperatures in an all-glass
apparatus, following the method described in Chance et al. (Chance et al., 2010). We initially filled the absorption





glass vessel with 20 ml n-hexane (99.95%, Merck), and then weighed it to determine the combined mass. We then
immersed the absorption vessel into a wide-necked Dewar vessel, filled with an acetone/dry ice mixture (at −80 ±
3 °C). After temperature equilibration, the $I_2$ molecules, carried by 50 mlpm $N_2$ flow from the permeation device, were
bubbled through the absorption vessel. After a continuous collection for 5 hours, we removed the absorption apparatus
from the cooling mixture, and allowed it to warm to room temperature prior to disassembling the setup to prevent any
losses of iodine on the tip of the inlet capillary. The absorption vessel was then re-weighed; the mass compared with
that prior to absorption was less than 2%, indicating a negligible loss during the trapping process. The $I_2$ / n-hexane
sample solutions were stored at 4 °C for 14 hours before being subjected to analysis.

We determined the $I_2$ concentration of the samples using a UV/Vis spectrophotometer (Shimadzu Model UV2450) at
a wavelength of 522 nm. We established a calibration curve via a set of $I_2$ solutions ranging from 270 to 5300 nmol,
diluted with n-hexane from a freshly prepared stock solution (0.5 g $L^{-1}$). Repetition of the same analysis after 2 and 7
days yielded identical results, confirming that the sample solutions were stable at 4 °C. As an alternative analytical
approach, we also quantified the $I_2$ concentration in the sample solutions using an inductively coupled plasma mass
spectrometer (ICP-MS, Agilent 7800). Before introducing to the ICP-MS, the sample solutions were treated with
$NaHSO_3$ water solution (0.100 M), accomplishing efficient hexane-to-water extraction and simultaneous reduction of
iodine to iodide (Schwehr et al., 2005) (Agilent Clinical Sample Preparation Guide (v3), *ref.* ISO 17294-2). The ICP-
MS results were in good agreement with those from the UV/Vis spectrophotometry.

We conducted the $I_2$ trapping and quantification experiments in triplicate with satisfactory reproducibility (standard
deviation < 10%). The calculated iodine permeation rate at 50 mlpm $N_2$ flow and 140 °C oven temperature is 278 ±
12 ng $min^{-1}$ (mean ± standard deviation). This result was used as the benchmark to estimate temperature-corrected
permeation rates according to the formula provided by the permeation tube vendor (VICI Metronic). We checked the
validity of the temperature-corrected values by conducting a second iodine absorption experiment in which the iodine
permeation tube was kept at 130 °C with an $N_2$ flow rate of 50 mlpm, and the determined permeation rate agreed
within 10% of the calculated value. We then diluted the $I_2$ flow to seven different values and measured the flow with
the Br-MION-CIMS. We repeated the calibration five times; we show the data along with a linear fit between the $I_2$
concentration and normalized $I_2$ signal in Fig. 4 (a). The slope of the line gives a calibration coefficient of $6.3 \times 10^{10}$
molec $cm^{-3}$ per normalized signal (cps $cps^{-1}$), with $R^2$ of 0.98 and an overall uncertainty of ± 45%.

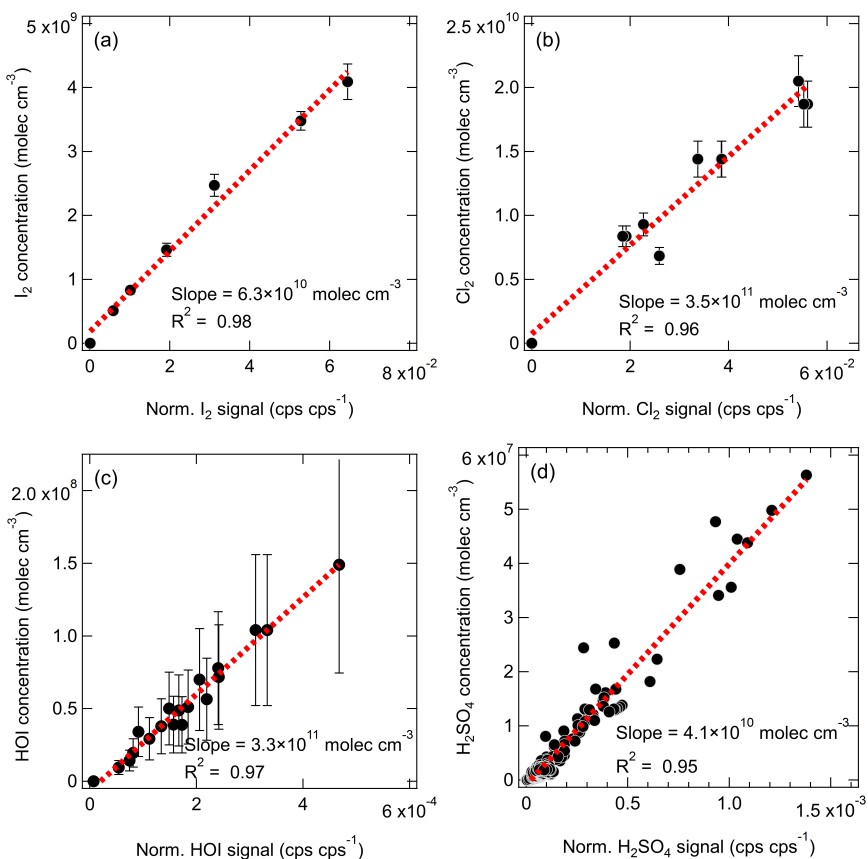

**Figure 4.** The absolute concentrations (molec cm$^{-3}$) *vs.* the normalized signals (cps cps$^{-1}$) measured with the Br-MION-CIMS for (a) I$_2$, (b) Cl$_2$, (c) HOI, and (d) H$_2$SO$_4$. The red dashed lines are the linear fittings. The overall factor 2 systematic scale uncertainty on [H$_2$SO$_4$] is not shown here.

### 3.3.3 Cl$_2$ calibration using a permeation tube

We used a commercial chlorine permeation tube (VICI Metronic) as a source for offline calibration. We passed a 20 mlpm high-purity nitrogen (99.999%) flow at room temperature through a 25 cm long stainless-steel tube (½" O.D.) containing the permeation tube. We quantified the permeation rate of Cl$_2$ following a procedure described in a previous study (Finley and Saltzman, 2008). The output of 20 mlpm flow was bubbled into a buffered aqueous potassium iodide solution (2.0 % KI (*m/v*), prepared in 1.00 mM aqueous phosphate buffer, pH = 7.0) filled in an all-glass two-stage serial absorption apparatus (stage 1 = 100 ml; stage 2 = 50 ml) for 3 hours and kept at room temperature. The Cl$_2$ oxidized the iodide (I$^-$) into iodine (I$_2$) once contacting with the KI absorption solution, and the I$_2$ further reacted with the excess KI present in the absorption solution to form I$_3^-$, which can be quantified by UV/Vis-spectrophotometry. We analyzed the resulting sample solutions with an UV/Vis spectrophotometer (Shimadzu Model UV-1800) using 1-cm quartz cells at 352 nm, corresponding to the I$_3^-$. We detected no I$_3^-$ in the second stage absorption solution, indicating that all the chlorine was quantitatively trapped and rapidly converted to I$_3^-$ within the first absorption unit.



The samples were quantified relative to $I_3^-$ standards in the range of 5 to $68 \times 10^{-6}$ M, prepared by dilution of a stock
obtained by dissolving 174 mg iodine in 200 ml of a solution containing 2 % KI in 1.00 mM aqueous phosphate buffer,
pH 7.0. From this calibration curve, we calculated a molar absorptivity of 26,800 L mol$^{-1}$ cm$^{-1}$, which is consistent
with the values reported in the literature (Finley and Saltzman, 2008; Kazantseva et al., 2002). Samples and standard
solutions were re-analyzed after being stored in the dark at room temperature for 24 hours, and the results were within
3 % of those obtained with the fresh solutions. We repeated the absorption experiment, and the calculated chlorine
permeation rate at room temperature was $764 \pm 74$ ng Cl$_2$ min$^{-1}$ (mean $\pm$ standard deviation).

The Cl$_2$ permeation source was run continuously for 12 hours prior to calibration experiments to ensure complete
system equilibrium. A two-stage dilution system similar to the setup of (Gallagher et al., 1997) was set up for diluting
the output of the Cl$_2$ permeation device. The 20 mlpm of N$_2$ stream emerging from the Cl$_2$ permeation device (operated
at room temperature) was diluted in a stream of 6 slpm of dry N$_2$. Then, a small fraction of this mixture (50 to 300
mlpm) was further mixed with the total flow of 25 slpm of N$_2$ (20 slpm dry N$_2$ + 5 slpm humidified N$_2$) before being
sampled by the Br-MION-CIMS. The calibration coefficient for Cl$_2$ was determined to be $3.5 \times 10^{11}$ molec cm$^{-3}$ per
normalized signal (cps cps$^{-1}$) from three separate calibration experiments (Fig 4 (b)), with an accuracy of 30 %.

**3.3.4 HOI calibration using a calibrator**
We produced a continuous HOI source via the reaction of I$_2$ and hydroxyl radical (OH) in a setup similar to the sulfuric
acid (H$_2$SO$_4$) calibrator (Kürten et al., 2012). The OH was generated by photolyzing H$_2$O with a mercury (Hg) lamp
at 184.9 nm, whose calibrated intensity was used to estimate the OH concentration. We tested the system by removing
the I$_2$ or OH source from the calibrator, upon which HOI production was undetectable, confirming that any single
reactant did not produce HOI. A numerical model was constructed to predict the mean HOI concentration entering the
Br-MION-CIMS, which is analogous to the model used for H$_2$SO$_4$ calibration (Kürten et al., 2012). We only included
the formation pathway of I$_2$ + OH to HOI in the model for simplicity; the other pathway of IO + HO$_2$ was considered
minor as IO forms at a relatively slow rate via the reaction of I radical and O$_3$. We produced a range of HOI
concentrations by varying I$_2$ and OH concentrations in the calibrator. We show in Fig. 4 (c) the linear correlation
between the modeled HOI concentrations and measured HOI signals. The slope of the fit line corresponds to a
calibration coefficient of $3.3 \times 10^{11}$ molec cm$^{-3}$ per normalized signal (cps cps$^{-1}$), with an overall uncertainty of $\pm$ 55%.
The good correlation (R$^2$ = 0.97) including various H$_2$O levels also indicates that H$_2$O concentrations did not affect
the HOI detection.

**3.3.5 H$_2$SO$_4$ calibration using a nitrate-CIMS**
We derive the H$_2$SO$_4$ calibration coefficient for the Br-MION-CIMS using the absolute H$_2$SO$_4$ concentrations
measured with a pre-calibrated nitrate-CIMS. The calibration protocol of H$_2$SO$_4$ in the nitrate-CIMS has been
described in detail previously (Kürten et al., 2012). The H$_2$SO$_4$ time series used for the inter-method calibration covers
a wide concentration range from less than $5.0 \times 10^4$ (detection limit of the nitrate-CIMS) to $6.0 \times 10^7$ molec cm$^{-3}$. For
Br-MION-CIMS, although both HSO$_4^-$ and H$_2$SO$_4$·Br$^-$ appear as distinct peaks for sulfuric acid, we only use the



normalized $H_2SO_4 \cdot {}^{79}Br^-$ for the inter-calibration, as $HSO_4^-$ (m/z = 96.96) has substantial interference from the reagent
ion $H_2O \cdot {}^{79}Br^-$ (m/z = 96.93). We show in Fig. 4 (d) the linear fit ($[H_2SO_4] = 4.1 \times 10^{10}$ molec cm$^{-3} \times H_2SO_4 \cdot {}^{79}Br^-/({}^{79}Br^-$
$+ H_2O \cdot {}^{79}Br^-) - 9.3 \times 10^5$) between the two $H_2SO_4$ traces with a correlation coefficient of 0.95. The calculated $H_2SO_4$
calibration coefficient is $4.1 \times 10^{10}$ molec cm$^{-3}$ per normalized signal (cps cps$^{-1}$).

**3.3.6 Connecting sensitivity to binding enthalpy**
Beyond the species for which we carried out calibrations, there are many more, especially iodine species, that cannot
be directly calibrated due to a lack of authentic standards or generation methods. However, the sensitivity of an iodide-
CIMS towards analytes can be predicted by the cluster binding enthalpy, calculated by relatively simple quantum
chemical methods (Iyer et al., 2016). This holds for the bromide-CIMS as well. In the instrument, ion clusters, formed
from reactions between analytes and reagent ions, are guided and focused by ion optics during transmission to the
detector. The electric forces applied to the clusters enhance their collision energies with carrier gas molecules. If
sufficient energy is transferred during the collision, cluster fragmentation may occur, affecting the instrument
sensitivity for the analytes (Passananti et al., 2019). However, clusters with higher binding enthalpy will be more
easily preserved and detected. Analytes that bind to the reagent ions with enthalpies higher than a critical level are
likely detected at maximum sensitivity (kinetic-limited detection) by the instrument. For example, the calculated
critical enthalpy is 25 kcal mol$^{-1}$ for the iodide-CIMS used in Iyer et al. (Iyer et al., 2016) and Lopez-Hilfiker et al.
(Lopez-Hilfiker et al., 2016), calculated at the DLPNO-CCSD(T)/def2-QZVPP//PBE/aug-cc-pVTZ-PP level of theory.

For the bromide chemical ionization, there are two types of fragmentation pathways:
1)    Reversion to the original form of $Br^-$ and analyte
$$X\text{-}H \cdot Br^- \rightarrow X\text{-}H + Br^- \qquad\qquad (1)$$
2)    Proton transfer from the analyte to $Br^-$
$$X\text{-}H \cdot Br^- \rightarrow X^- + HBr \qquad\qquad (2)$$
where the X-H is the hydrogen bond donor. An analyte may be expected to be detected at the maximum sensitivity
when the dissociation enthalpy for the first pathway is either a) much higher than the critical enthalpy (dissociation
back to the reactants does not occur), or b) lower than the critical enthalpy, but much higher than that of the second
pathway (dissociation back to the reactants would occur, but it is not competitive with the other dissociation channel).
Whether the enthalpy for the second pathway is higher than the critical enthalpy does not directly affect the sensitivity,
as both $X\text{-}H \cdot Br^-$ and $X^-$ can be measured and counted. Taking $H_2SO_4$ as an example, the dissociation enthalpies for
the first and second pathways are 41.1 and 27.9 kcal mol$^{-1}$, respectively. If some of the $H_2SO_4 \cdot Br^-$ dissociate, they
preferably become $HSO_4^-$ and are detectable by the Br-CIMS. We list the cluster formation enthalpies for a selection
of halogen containing species in Table 1 and the corresponding cluster dissociation enthalpies in Table 2.

While we were unable to experimentally establish a correlation between sensitivities and binding enthalpies due to
limited quantifiable halogen species, we can predict the tentative critical enthalpy as the binding enthalpy of a species
that is likely detected at the maximum sensitivity. Among all the calibration coefficients listed in Table 3, $H_2SO_4$ and





$I_2$ have the lowest calibration coefficients (highest sensitivities); and their coefficients are almost the same for both
the online and offline calibrations, with discrepancies well within the systematic uncertainties. Thereby, we conclude
that both $H_2SO_4$ and $I_2$ are detected at the maximum sensitivity, suggesting a critical enthalpy not higher than 33.7
kcal mol$^{-1}$. We can then infer the sensitivity for other species that are difficult to calibrate by comparing their binding
enthalpies to those of the benchmark species. For example, ICl and IBr should have the maximum sensitivity, since
the dissociation enthalpies for $ICl \cdot Br^-$ and $IBr \cdot Br^-$ are both much higher than 33.7 kcal mol$^{-1}$ (Table 2). Although
$HIO_3 \cdot Br^-$ has a lower dissociation enthalpy than the critical value, the favored dissociation pathway is proton transfer
(the second pathway); $HIO_3$ can thus be considered as a maximum sensitivity species detectable as $IO_3^-$ ions after
proton transfer. This is consistent with the fact that both $HIO_3 \cdot Br^-$ and $IO_3^-$ are detected in Figure 1, so is the case with
$H_2SO_4$. We thus assume that $HIO_3$ has a kinetic calibration coefficient of $3.8 \times 10^{10}$ molec cm$^{-3}$ cps cps$^{-1}$, the value for
$H_2SO_4$. However, the lowest dissociation enthalpies of $HOI \cdot Br^-$ and $Cl_2 \cdot Br^-$ are 26.9 and 22.3 kcal mol$^{-1}$, respectively,
consistent with their higher calibration coefficients of $3.3 \times 10^{11}$ and $3.5 \times 10^{11}$ molec cm$^{-3}$ cps cps$^{-1}$. The dissociation
enthalpies for $IO \cdot Br^-$, $OIO \cdot Br^-$, and $HIO_2 \cdot Br^-$ are 24.5, 23.2, and 29.2 kcal mol$^{-1}$, respectively.
We would expect that their sensitives are lower than the maximum sensitivity. Since the dissociation enthalpies for $IO \cdot Br^-$ and $OIO \cdot Br^-$ are
between those of $HOI \cdot Br^-$ and $Cl_2 \cdot Br^-$, a similar calibration coefficient of around $3.0 \times 10^{11}$ molec cm$^{-3}$ cps cps$^{-1}$ could
be applied.
Further, we estimate the detection limit of the calibrated species. The detection limit is defined as the analyte
concentration, corresponding to the sum of the mean signal and three times the standard deviations (3σ) of the
background fluctuations during a two-hour background measurement. We derive the detection limit of HOI, $HIO_3$, $I_2$,
and $H_2SO_4$ to be $5.8 \times 10^6$, $1.2 \times 10^5$, $6.3 \times 10^5$, and $2.0 \times 10^5$ molec cm$^{-3}$ (or 0.2, 0.005, 0.03, and 0.008 pptv), respectively,
for a 2-min integration time.
**3.4 Comparison between Br-MION-CIMS and Br-FIGAERO-CIMS**
While Br-MION-CIMS and Br-FIGAERO-CIMS use the same chemical ionization scheme, their designs differ in the
ion-molecule reaction chamber (IMR). MION is an atmospheric pressure (1 bar) drift tube; analyte molecules gain an
electric charge in an axial laminar flow. FIGAERO is connected to a cone-shaped IMR chamber operated at a reduced
pressure (150 mbar); the sample flow is injected into the inlet via an orifice, necessarily causing turbulence and wall
interactions in the IMR region. The atmospheric pressure and reduced pressure IMRs are both widely used for trace
gas measurements. We thus compare iodine species measurements from Br-MION-CIMS and Br-FIGAERO-CIMS,
to better understand the performance and applicability of bromide ionization scheme.
**3.4.1 Signal trend and detection limit**
We show in Fig. 5 the same new-particle formation event as in Fig. 2, to illustrate the time series for $HIO_3 \cdot Br^-$, $HOI \cdot Br^-$,
$IO \cdot Br^-$, and $I_2 \cdot Br^-$, measured with Br-MION-CIMS (red circles) and Br-FIGAERO-CIMS (grey sticks), respectively.
Note that the FIGAERO alternates between gas and particle measurements; here we show only the gas-phase signals.
Clear and concurrent signals of $HIO_3$, HOI, IO, and $I_2$ are evident from both the Br-MION-CIMS and Br-FIGAERO-





CIMS. Prior to the NPF event (08:11), there was no photochemical production and thus virtually no signal of oxidized
iodine species in both instruments. The dark reaction of ozone with $I_2$ did not proceed at a significant rate, due to the
low rate coefficient and low levels of $I_2$. Signals detected during this period are considered as the persistent background,
coming from electronic noise or other sources such as the ionizer, carrier flows, or long-term "memory" in the case
of the Br-FIGAERO-CIMS. Not surprisingly, the Br-MION-CIMS has a near-zero background for all analytes. For
$HIO_3$ (Fig. 5 (a)), the background signal in the Br-FIGAERO-CIMS is also negligible; however, IO shows a substantial
persistent background (Fig. 5 (c)) in the Br-FIGAERO-CIMS. After the NPF event (10:21), the photochemical
production of oxidized iodine species was terminated and vapor concentrations decayed exponentially due to dilution
and losses to chamber walls. The $I_2$ signal increases after the event termination because it is no longer photolyzed.

Applying the calibration coefficients, we convert the Br-MION-CIMS signals to absolute concentrations, and
subsequently correlate them with signals measured with Br-FIGAERO-CIMS. We then estimate the tentative detection
limits for HOI and $HIO_3$ in the Br-FIGAERO-CIMS to be $3.3 \times 10^7$ and $5.1 \times 10^6$ molec cm$^{-3}$ (versus $5.8 \times 10^6$ and $1.2 \times 10^5$
molec cm$^{-3}$ in the Br-MION-CIMS), respectively, at $3\sigma$ of the background signal for a 2-min integration time during
a two-hour period; they are in general one order of magnitude higher than those in the Br-MION-CIMS. This is in line
with the higher background signals observed in the Br-FIGAERO-CIMS. We are unable to estimate the $I_2$ detection
limit in the Br-FIGAREO-CIMS, due to a lack of $I_2$ background measurement; but Br-FIGAERO-CIMS can and did
detect $I_2$ at the low pptv level with good fidelity.

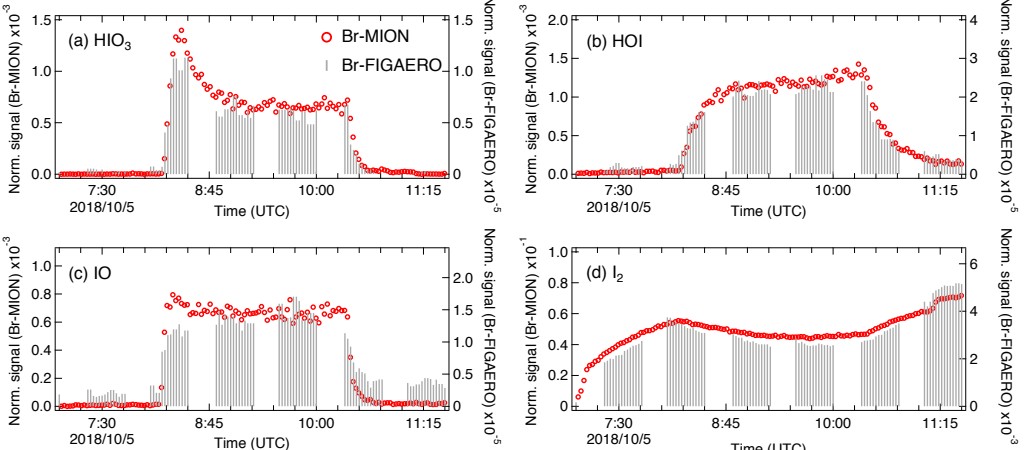

**Figure 5.** Signal comparison of selected iodine species measured with Br-MION-CIMS and (gas-phase) Br-FIGAERO-CIMS,
respectively, during the same iodine new-particle formation experiment shown in Fig. 2.

**3.4.2 Rise and decay time constants**
In order to quantitatively compare the performance of the two types of IMRs, we set the initiation (08:11) and
termination (10:21) of NPF event as $t = 0$, and fit the rise and decay rates of $HIO_3$, HOI and IO in both instruments,
respectively.



When we initiated the photochemistry, oxidized iodine species rapidly built up toward an asymptote; we thus fit their
time-series signals individually with a four-parameter sigmoid function using a least-squares fitting algorithm:
$$S_i(t) = a + (b - a)/\left(1 + e^{(-(t-t_{app})/c)}\right) \qquad (3)$$

where $a$ and $b$ represent the background and asymptotic value of the sigmoid function respectively, $c$ is the exponential
time constant of the signal change, and $t_{app}$ is the time at which the 50 % value between plateau and background is
reached (50 % appearance time). As we were unable to separate the time scale of chemical reactions from that of
instrument response, we use the time constants to represent the overall response for the purpose of comparing
instrument performance.

We show in Fig. 6 (a), (c), and (e) the rise rate fits of HIO₃, HOI, and IO, respectively. HIO₃ signals rose with a time
constant of 102 s in the Br-MION-CIMS and 108 s in the Br-FIGAERO-CIMS, both with $t_{app}= \sim 300$ s. The fitted time
constants of HOI are slightly longer than those of HIO₃, with 120 s in the Br-MION-CIMS and 114 s in the Br-
FIGAERO-CIMS. IO signals stabilized the earliest, thus have the fastest time constants of 48 s in the Br-MION-CIMS
and 84 s in the Br-FIGAERO-CIMS. The instrumental differences are small for HIO₃ and HOI, but larger for IO. When
colliding with the IMR surface, HIO₃ condenses irreversibly; it thus makes sense that the Br-MION-CIMS and Br-
FIGAERO-CIMS signals show the same time constant for HIO₃. Semi-volatile HOI, however, can return to the gas
phase from the walls depending on the surface coverage of HOI and the vapor concentration. Additionally, the
heterogeneous reaction of aqueous iodide (I⁻) and ozone (Carpenter et al., 2013) could also contribute to the emission
of HOI from the IMR wall in the FIGAERO. As the evaporation flux is typically a function of the amount analyte on
the surface, the buffering effect could degrade the instrument time response upon changes in analyte concentration.
Here, however, we did not observe a significant memory effect, likely because the HOI concentration was too low to
fully saturate the IMR surface or because any HOI evaporation was suppressed due to an enhanced accommodation
coefficient of HOI on the metal surface. We expect IO to be prone to loss on the metal surface due to its radical nature.

We also fit the exponential decay time constants of these iodine species to test this interpretation (Fig. 6 (b), (d), and
(f)). After photochemistry was terminated at the end of the NPF event, only two sinks drove the vapor concentration
decay – dilution and wall loss. Memory effects could also influence the signal time constant. The dilution loss rate
was around $2.1\times10^{-4}$ s⁻¹ (4760 s time constant) for all species in the chamber, determined by the total chamber flow
rate and the chamber volume. Wall loss rates, however, vary for species with different diffusion constants. The decay
rates of HIO₃ are 400 s for the Br-MION-CIMS and 370 s for the Br-FIGAERO-CIMS, much faster than the dilution
loss. For comparison, the time constant for H₂SO₄ vapor loss was 300 s. These time constants are thus consistent with
wall loss (around $2.2\times10^{-3}$ s⁻¹). The IO decay time constant is 294 s for the Br-MION-CIMS and 435 s for the Br-
FIGAERO-CIMS. The time constant for the Br-MION-CIMS indicates that the decay of IO is also driven by wall loss,
so the net flux during this period was thus towards the wall rather than from the wall. Therefore, the difference of IO
between instruments may well be attributed to the persistent background from the ionizer of the FIGAERO. The HOI





signals have longer decay time constants in both instruments of 909 s for the Br-MION-CIMS and 714 s for the Br-
FIGAERO-CIMS; this may reflect a time constant for depletion of HOI adsorbed to the chamber walls.

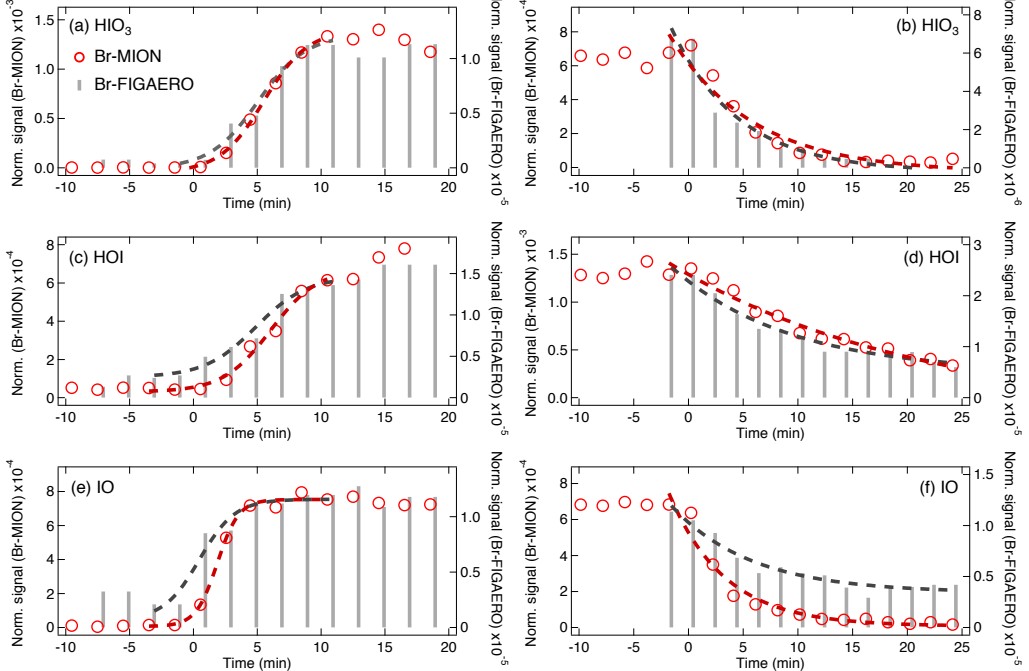

**Figure 6.** Rising ((a), (c), and (e)) and decay ((b), (d), and (f)) rate comparison of selected iodine species measured with Br-MION-
CIMS and Br-FIGAERO-CIMS, respectively, during the same iodine new-particle formation event in Fig. 2 and 5.

**4 Summary and conclusion**
We confirm in this study that bromide chemical ionization is a suitable technique for the time-resolved, highly
sensitive, and simultaneous measurements of iodine species and sulfuric acid. The Br-MION-CIMS shows constant
sensitivity throughout the relative humidity range of 40 to 80 % at -10 °C, as long as the sum of the two reagent ions
($Br^-$ + $H_2O \cdot Br^-$) is used for signal normalization. This demonstrates the applicability of this technique to field
measurements in the ambient marine environment.

We quantify iodine species and sulfuric acid via offline calibrations (i.e. permeation tube and calibrator) and inter-
method calibrations (i.e. CE-DOAS and pre-calibrated nitrate-CIMS); different methods result in consistent
calibration coefficients. Further, we calculate the binding enthalpies between the calibrated species and reagent ions,
which qualitatively agree with the corresponding calibration coefficients. This indicates that the quantum chemical
calculations can be employed along with the calibration experiments to determine the sensitivities for unquantifiable
species; more work is required to further establish the correlation between calibration coefficients and binding
enthalpies.





Further, using inter-method and offline calibrations, we estimate the detection limits of HOI, $HIO_3$, $I_2$, and $H_2SO_4$ in
Br-MION-CIMS being $5.8 \times 10^6$, $1.2 \times 10^5$, $6.3 \times 10^5$, and $2.0 \times 10^5$ molec $cm^{-3}$, respectively, for a 2-min integration time
during a two-hour period. To our knowledge, the simultaneous measurements of various iodine species and sulfuric
acid with low detection limits are unprecedented for online techniques. Detection limits for HOI and $HIO_3$ in the Br-
FIGAERO-CIMS are $3.3 \times 10^7$ and $5.1 \times 10^6$ molec $cm^{-3}$, which are in general one order of magnitude higher than those
in the Br-MION-CIMS. The signal comparison between the two instruments also shows that the Br-CIMS can be
coupled to both the atmospheric pressure and the reduced pressure interfaces for iodine species and sulfuric acid
measurements in the marine environment.


*Data availability.* Data available on request from the authors.

*Author Contributions.* M.W., X.-C.H., Y.-J.T and H.F. wrote the manuscript. X.-C.H., Y.-J.T., M.W. and M.Sip.
designed the experiments. X.-C.H., Y.-J.T. and J.S. carried out the Br-MION-CIMS measurements, M.W., D.C. and
V.H. carried out the Br-FIGAERO-CIMS measurements, and H.F. carried out the CE-DOAS measurements. Y.-J.T.,
X.-C.H., H.F., D.C., J.S. and M.Sim. performed the calibrations. S.I., X.-C.H. and T.K. carried out the quantum
chemical calculations. M.W. performed the comparison analysis of the Br-FIGAERO-CIMS and Br-MION-CIMS.
N.-M.D., T.K., M.R., R.V and M.Sip. commented on the manuscript. All other co-authors participated in either the
development and preparations of the CLOUD facility and the instruments, and/or collecting and analyzing the data.

*Competing interests.* The authors declare that they have no conflict of interest.

*Acknowledgement.* We thank the European Organization for Nuclear Research (CERN) for supporting CLOUD with
important technical and financial resources and for providing a particle beam from the CERN Proton Synchrotron.
We also thank Juhani Virkanen and Heini Ali-Kovero for providing assistance in the laboratory analytical experiments.

*Financial support.* This research has received funding from the US National Science Foundation (AGS-1531284,
AGS-1801574 and AGS-1801280), Academy of Finland (projects: 296628, 328290, Centre of Excellence 1118615)
and the European Research Council (ERC) under the European Union's Horizon 2020 research and innovation
programme (GASPARCON, grant agreement no. 714621). The FIGAERO-CIMS was supported by an MRI grant for
the US NSF AGS-1531284 as well as the Wallace Research Foundation.

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



**Table 1: Cluster formation enthalpies of different species with bromide ion.** The cluster geometries are optimized
at the ωB97X-D/aug-cc-pVTZ-PP level at 298.15 K. The enthalpies are calculated at the DLPNO-CCSD(T)/def2-
QZVPP//ωB97xD/aug-cc-pVTZ-PP level at 298.15 K.

| Cluster formation pathway | Formation enthalpies (kcal mol$^{-1}$) |
|---|---|
| $Cl_2 + Br^- \rightarrow Cl_2 \cdot Br^-$ | -22.3 |
| $OIO + Br^- \rightarrow OIO \cdot Br^-$ | -23.2 |
| $IO + Br^- \rightarrow IO \cdot Br^-$ | -24.5 |
| $HIO_3 + Br^- \rightarrow HIO_3 \cdot Br^-$ | -26.6 |
| $HOI + Br^- \rightarrow HOI \cdot Br^-$ | -26.9 |
| $HIO_2 + Br^- \rightarrow HIO_2 \cdot Br^-$ | -29.2 |
| $I_2 + Br^- \rightarrow I_2 \cdot Br^-$ | -33.7 |
| $ICl + Br^- \rightarrow ICl \cdot Br^-$ | -33.8 |
| $IBr + Br^- \rightarrow IBr \cdot Br^-$ | -36.7 |
| $H_2SO_4 + Br^- \rightarrow H_2SO_4 \cdot Br^-$ | -41.1 |
| $I_2O_4 + Br^- \rightarrow I_2O_4 \cdot Br^-$ | -42.6 |
| $I_2O_5 + Br^- \rightarrow I_2O_5 \cdot Br^-$ | -53.2 |


**Table 2: Fragmentation reaction enthalpies of different species with bromide ion.** The cluster geometries are
optimized at the ωB97X-D/aug-cc-pVTZ-PP level at 298.15 K. The enthalpies are calculated at the DLPNO-
CCSD(T)/def2-QZVPP//ωB97xD/aug-cc-pVTZ-PP level at 298.15 K.

| Cluster fragmentation pathway | Fragmentation enthalpies (kcal mol$^{-1}$) |
|---|---|
| $Cl_2 \cdot Br^- \rightarrow Cl_2 + Br^-$ | 22.3 |
| $Cl_2 \cdot Br^- \rightarrow BrCl + Cl^-$ | 22.3 |
| $HIO_3 \cdot Br^- \rightarrow HIO_3 + Br^-$ | 26.6 |
| $HIO_3 \cdot Br^- \rightarrow IO_3^- + HBr$ | 20.8 |
| $HIO_3 \cdot Br^- \rightarrow IO_2^- + HOBr$ | 52.0 |
| $HOI \cdot Br^- \rightarrow HOI + Br^-$ | 26.9 |
| $HOI \cdot Br^- \rightarrow IO^- + HBr$ | 57.7 |
| $HOI \cdot Br^- \rightarrow I^- + HOBr$ | 31.3 |
| $HIO_2 \cdot Br^- \rightarrow HIO_2 + Br^-$ | 29.2 |
| $HIO_2 \cdot Br^- \rightarrow IO_2^- + HBr$ | 43.8 |
| $HIO_2 \cdot Br^- \rightarrow IO^- + HOBr$ | 42.2 |
| $I_2 \cdot Br^- \rightarrow I_2 + Br^-$ | 33.7 |





| | |
|---|---|
| $I_2 \cdot Br^- \rightarrow IBr + I^-$ | 33.8 |
| $ICl \cdot Br^- \rightarrow ICl + Br^-$ | 33.8 |
| $ICl \cdot Br^- \rightarrow IBr + Cl^-$ | 39.8 |
| $ICl \cdot Br^- \rightarrow BrCl + I^-$ | 42.0 |
| $IBr \cdot Br^- \rightarrow IBr + Br^-$ | 36.7 |
| $IBr \cdot Br^- \rightarrow Br_2 + I^-$ | 39.4 |
| $H_2SO_4 \cdot Br^- \rightarrow H_2SO_4 + Br^-$ | 41.1 |
| $H_2SO_4 \cdot Br^- \rightarrow HSO_4^- + HBr$ | 27.9 |


**Table 3: Calibration coefficients for selected species.**

| Species | Calibration coefficient (molec cm$^{-3}$ cps cps$^{-1}$) |
|---|---|
| $I_2$ | [a] $2.7 \times 10^{10}$ |
| $I_2$ | [b] $6.3 \times 10^{10}$ |
| $Cl_2$ | [b] $3.5 \times 10^{11}$ |
| HOI | [b] $3.3 \times 10^{11}$ |
| $H_2SO_4$ | [a] $4.1 \times 10^{10}$ |

a: inter-method calibrations
b: offline calibrations