# Peer review of "Measurement of iodine species and sulfuric acid using bromide chemical ionization mass spectrometers"

_Atmospheric Measurement Techniques, 2020_

## Referee Comment (RC1) · Anonymous Referee #1 · 10 Feb 2021

**1   General Comments**

The authors present a study on the Bromine-MION CIMS, an impressive new tool to selectively and extremely sensitively measure several relevant iodine species and sulfuric acid at the CLOUD chamber with clearly the same potential for atmospheric measurements. The combination of an extremely clean chemical ionization procedure with a very high-resolution time-of-flight mass spectrometer renders unprecedented selectivity and obviously sensitivity which is shown by means of instrument intercomparisons and employing accurate absolute calibration techniques as well as a rough method based on intercomparisons of measured and ab-initio calculated cluster-ion formation

and dissociation enthalpies. Overall this is a very relevant study fitting perfectly into the scope of AMT.

However, the way the involved experiments are presented and discussed needs considerable revisions in order to enhance the usefulness to the CIMS community and also be comprehensible for non-specialist readers. I give my general concerns in the following paragraphs and many detailed comments and proposals in the next section.

In general the manuscript is written in a way hard to follow for non-specialists in the CIMS technique. Just as one example the widely used ion signal ratios should be briefly rationalized in terms of ionization reaction rates before being discussed. Although the paper refers to the relevant instrument papers basic procedures and derived quantities etc. used in 'every day live' by the authors must be briefly explained. The reader can not be expected to look up 10 other papers in order to properly follow the basic course of the manuscript.

In places sloppy and unclear formulations are used, e.g. 'uncertainty' instead of clearly stating if accuracy or precision are referred to and if 1 or 2 standard deviations are given. Also details of the experiments need to be much better defined where the relevant parameters have clear influence on the results presented. As an example critical settings of electrical fields employed in the first transfer stages which have strong impact on the relative signals of ion clusters and bare ions in the mass spectra are not even mentioned. Reproducability of the results presented is not given if the reader has to try to isolate such relevant information from the papers cited and can't be sure that they have not been changed. As another example it is not even mentioned which type of mass analyzer is used for the MION-CIMS or also the FIGAERO and what the relevant mass resolutions are. Therefore it needs to be clearly stated in the methodology section 2 in which relevant operational conditions the instrument(s) have been operated. Pressures and flows employed in the IMR and $1^{st}$ transfer stage and respective reaction times (as well as the electric fields applied in the ion transfer) should be given in the experimental section and not implicitely towards the end of section 3! This is

absolutely neccessary for sake of reproducability. A table may do the job here.

In places purely qualitative statements in the discussion should be quantified by giving the relevant parameters such as for relative reaction rates etc. (see in the specific comments).

For the $I_2$ and $Cl_2$ permeation devices which have been nicely qualified I am wondering why no parallel gravimetric control has been done as this is a very simple and reliable way of calibrating devices with such compounds and high permeation rates approaching the mg/day regime.

The operations around the permeation devices have been worked out in quite some detail although these procedures are not new but have also been described in earlier publications which are cited. I don't criticize that detail, however, there is a strong misbalance to other experimental details especially since this part has been placed into the results and discussion section 3. Here certainly some restructuring is neccessary.

Two independent approaches for the $I_2$ calibration have been employed and presented. Both look very good but differ by more than a factor two from the CE-DOAS intercomparison to the permeation tube approach most obvious in Table 3. This is not clearly mentioned nor discussed in the manuscript. On the contrary the summary states that 'different methods result in consistent calibration constants'!? Since both methods used are in principle absolute techniques and individual errors seem much lower there must be some conclusion on this discrepancy seemingly outside the combined error bars. Has the DOAS system been calibrated directly employing the permeation device in order to resolve this? This point clearly needs revision and clarification!

An approach to compare the performance of the MION and FIGAERO instruments especially with respect to surface effects is presented in section 3.4.2 on rise and decay time constants. The tentative interpretation presented seems reasonable, however my feeling here is that the fitting analysis does not give any significant additional information on the signal evolution than obvious already from the pure normalized signals.

[Figure]

MION and FIGAERO agree well for $HIO_3$ and HOI. Therefore the evolutions probably indicate wall uptake and desorption processes possibly occuring somewhere from the chamber into the IMRs since else certainly much slower decays would be expected. The fitted evolutions just underline the instrument deviations which seem really significant only for IO with the obvious elevated background signals on the FIGAERO. This might also be due to some residual outgassing of the FIGAERO filters after the aerosol measuring phases which just ended before start and end of the photolysis. Another reason for the 'permanent' IO background in the FIGAERO might also be some unrecognized isobaric contaminant on the lower resolution (if I'm not mistaken) FIGAERO analyzer. However, in the current state of analysis I don't see any additional information provided by this analysis technique.

**2  Specific Comments**

**l.116:** Here an outline of a typical NPF experiment on the CLOUD chamber should be briefly presented with the operational details of the experiment, like flushing, I2 addition and control, etc as a function of experiment time. See general comments.

**l.130:** Details of the experiments need to be much better defined, see general comments

**l.166:** Accuracy is a defined quantity and I'm not aware of 'systematic accuracy'. Is 20% accuracy or rather precision meant here? For all statements on accuracy or precision it must be clearly mentioned which quantities are given, 1 or 2 standard deviations?

**l.171:** Isn't 'conformational sampling' the more common term?

**l.179:** The experimental details on the qualification of the $I_2$ and $Cl_2$ permeation devices and HOI calibrator operation from section 3 should be moved to here. Clearly these are methodological details. See general comments.

**l.187:** ... high mass resolution ... Here the mass analyser type and important specs should be detailed!

**l.222:** Does the RH range encompass all relevant experiments carried out? Lee et al., 2014, show that for I- ionization the major RH sensitivity of the calibration, especially for small organic acids, is found at quite low RH.

**l.228-232:** It may be worthwhile to although show the respective plot for the HOI normalization.

**l.246:** I propose to not present the rather lengthy equation in the text flow but as a separate equation.

**l.246:** Is there a physical motivation for the inclusion of the quadratic fit? If there is any relevance, apart from the good agreement with the linear relation, it should be clearly discussed. Else it can be neglected.

**l.255:** For this as well as the $Cl_2$ device all wetted materials used for the calibration gas generation should be detailed. Especially for the low flow rates used for flushing the permeation devices even compounds like the elemental halogens could be reduced due to surface effects.

**l.280:** Quantification would clearly help with this statement (e.g. ... agreement within XX% with ...)

**l.290:** The calibration constant reported for $I_2$ from the CE-DOAS intercomparison was $2.7E - 10 molec/cm^{-3}$, i.e. more than 60% lower as mentioned earlier. However, were does the 'overall uncertainty' (is this accuracy?) of 45% come from while the permeation rate is defined to better than 5%?

**l.334:** Give the relative reaction rate instead of ' ... relatively slow ...'

**l.351:** No statements on the accuracy of the $H_2SO_4$ measurements for the pre-calibrated nitrate-CIMS nor the Br-MION are given in this section. Just the caption of Fig.4 mentions a 'systematic scale uncertainty'. This is very annoying and makes me wonder if this has been left out on purpose ...

**l.348:** I propose to not present the rather lengthy equation in the text flow but as a separate equation.

**l.374:** The explanation is hard to comprehend for the non-specialist reader and rather leads to confusion. This should be improved upon e.g. by going step by step from the dissociation enthalpies via reaction rates etc.

**l.393ff:** Even though the derived calibration constants (and later detection limits) of the species not calibrated directly are quite crude they should be given in a table rather than in the text flow. I propose to include them into Table 3 as a third category c: derived from dissoc. enthalpies (or similar). Table 3 could be extended by another column for the det. limits for all species as well.

**l.421:** I find the use of 'NPF event' rather annoying since no particle data are shown or discussed, I propose to just refer to the 'photolysis interval' or similar.

**l.429:** Is there a continuous addition of $I_2so$ to keep it constant over the photolysis causing $I_2$ to increase when the lamps are off?

**3  Figures and captions**

**Fig.1:** I really do like the figure, however, the caption will have to be extended considerably in order to properly inform and guide the reader! First there should be 2

defined panels (upper and lower, a) and b), ...). Mass defect could be defined in the text so not to extend the caption. More clear identification of the experiment (date, ...) may not be absolutely neccessary here but still enhance transparency. Upper panel: symbol area is proportional to the logarithm of the signal AREA which has a unit of counts/s. However, then the y-axes in the lower panel can not have the same unit! Please, align these details!

Lower panel: The explanations given in the upper left spectrum must be detailed. Although obvious it should be indicated that colors in the panels do match. It might be sensible to use the same x-scaling for all spectra.

What is the averaging time of the spectra shown?

**Fig.2:** The time series plot shows peak area (not intensity) vs. time.

**Fig.3:** A different and zoomed color scale has to be selected here in order to better show the range of RH varied from just 40-80%.

Is there an explanation for the slight hockey stick shape visible at high RH and low $I_2$ concentrations? Is this a consequence of the DOAS det. limit?

**Fig.4:** The x-axis titles should give the detected ion clusters. The analyte species may be given in the caption.

**Fig.1:** abc

**4  Minor comments, typos, etc.**

**l.70:** I recommend to use 'absorption features' instead of cross sections.

**l.83:** The bromide ion ...

**l.114:** ... the atmospheric oxygen ratio of 0.21 ... (there aren't 79% of N2!)

**l.186:** ... bromide ionS ...

**l.247:** ... the whole campaign ... THE two curves ...

**l.327:** ... hydroxyl radicalS ...

**l.350:** This is a repetition from the equation given before ...

**l.414:** ... of THE bromide ...

**l.423:** ... and TO low levels of $I_2$... (The sentence may otherwise suggests that $I_2$ had not yet been added to the chamber at this point.)

**l.446:** ... of THE NPF event ...

**l.739, 743:** ... with bromide ionS.

---

## Referee Comment (RC2) · Anonymous Referee #2 · 24 Feb 2021

This is a well written and organized account of the application of a bromide cims instrument for the detection of various iodine containing species as well as sulfuric acid. The authors present nice account with much of the pertinent information related to their experiments. However, I find that there is much to be desired with respect to the errors associated with the methods used here and the various approximation used throughout the anaylsis. This is significant as the authors are presenting a very optimistic account of the ability of this method to detect extremely low levels of these species in a real-world marine environment. If in fact the authors can provide more details and support their conclusions through the additional information this is a truly remarkable performance and will be of wide interest to the field of atmospheric mass spectrometry. It is my opinion that more detail must be provided to support the publication of this manuscript in AMT and therefore recommend the following minor revisions to the manuscript.

I am having a difficult time understanding the true errors associated with your calibration factors presented throughout the manuscript. In some cases, you are calibrating using external calibration sources and in others you are calibrating using a secondary instrument for comparison. One issue with this whole description is the lack of information provided for exactly how these were performed. What I mean by this is how were the calibrations sources samples, through an inlet or directly into the IMR? Or when two instruments were compared were different inlets used between the two instruments. This is of particular importance when extrapolating the cluster binding energy approximated sensitivities to your instrumentation. I ask all of this because the actual true sensitivity to a given species is a combination of the ionization sensitivity as well as an instrument function which describes any inlet losses or differences in ionization potential between various instruments. The main point of your work is that this method can detect these species with the sensitivity and detection limits sufficient for real world marine environments, so this really matters here. How well do you actually know your sensitivities and detection limits? Are those an upper limits because they represent sensitivities without an inlet present? You have made many assumptions related to the calibration of your instrumentation and that is certainly not reflected in any robust error analysis that is provided to the reader.

On the topic of using cluster binding energies for the approximation of sensitivities, the authors suggest that because that sort of analysis has worked for iodide than it naturally works for the Br- ion chemistry, however, there is no evidence that this approach is true. Again this method is also complicated by the differing inlet response functions of the various species and would really only yield an upper limit to the ionization sensitivity not the true detection capabilities of a sampling system. For example, a binding energy does not take into account that sulfuric acid has a different transmission factor than IO

which would also vary with sampled ambient humidity on the inlet surfaces.

I think the calibration through instrument comparison method was used in this manuscript to transfer the MION calibrations to the FIGAERO, perhaps I incorrectly understood something. If that is the case, this issue of instrument functions becomes even more of a hindrance to the direct comparison of ambient observations because the two instruments operate at very different pressures with significantly different ionization schemes. These types of issues clearly need to be stated and incorporated into the stated performance of the instruments, preferably through a more robust error analysis. Although that may not be possible as the error in approximated calibration factors may not be available.

I would like to know more details on the what the background conditions were during the periods for which the detection limits were calculated. It is well known that when a CIMS instrument samples clean air the background values are continually reduced. In most real world applications conditions are far from clean and as such I would expect a relatively larger $H_2SO_4$ background than what the authors are stating. I am curious if this discrepancy comes from the fact that this instrument is sampling from what is perhaps the cleanest atmosphere available on earth which allows for unrealistic instrument backgrounds. Yes, that would be the functional instrument detection limits for sampling in the CLOUD chamber, however the results would not be directly translatable to a real-world environment where instrument background are likely significantly higher. If in fact the detection limits are determined from periods of chamber measurements, or even if the instrument is only sampling from the chamber and not room air then the authors should admit that this is really a best-case scenario for the stated detection limits. This would mean that real-world applications of this method in the marine atmosphere are unlikely to achieve these results. Unless of course the authors can support the idea that these instrument backgrounds are routinely achievable on a standard field deployable CIMS instrument. If this is the case data should be shown describing the various background and operating conditions. Also in general, during

the chamber experiments how were zeros performed?

The pressure at which these instrument are being run at leads me ask if the authors have identified any issues with the combination of Br- and O3 as it relates to secondary ion chemistry. It is fairly well known that the addition of O3 to an I- CIMS will result in the production of IO-, IO2-, and IO3- which can act as a secondary reagent ion even at low abundances. This presents challenges to interpretation of oxygen rich ions where (HIO2)Br- can actually be the result of a cluster of HOI with BrO-. The pressures used in this study are sufficiently high in an iodide CIMS to result in these reactions at this ozone level and I would encourage the authors to discuss this potential.

In figure 4 and elsewhere the sensitivities are given as a concentration over signal. This figure should be flipped as the signal in the instrument is dependent on the concentration not the other way around. It would be helpful if values for sensitivity were given in the convention standard for atmospheric mass spectrometry as signal/concentration.

I can summarize my main concern with respect to the authors not presenting the errors associated with the work appropriately with one example. There is a large discrepancy between the permeation source derived I2 sensitivity and the DOAS derived sensitivity 2.7e10 versus 6.3e10. That is a factor of 2 difference! It seems this is just glossed over and needs to be explained. These are even measured sensitivities with that large of a discrepancy so I am inclined to believe the errors in the approximations are at least as large and likely significantly larger than this difference.

Specific comments:

- Section 3.3.4 and 3.3.5 should be flipped as the first sentence of 3.3.4 states the calibration was performed as in 3.3.5

- Figure 2 caption doesn't seem to have any proton transfer reaction products as the caption would lead the reader to believe.

- Figure 6, as sulfuric acid is one of the main foci of this manuscript it really should be

included in this figure to show the response time of the measurement.

---

## Author Comment (AC1) · 8 Apr 2021

**RE: A point-by-point response to referee comments**

We thank the reviewer for dedicating time for a diligent review of the manuscript, and for providing valuable comments and detailed suggestions for improvements. We have carefully modified the manuscript to improve the legibility, to clarify critical instrument parameters, and to specify the potential limitation of this method in laboratory and field measurements. Below we provide a point-by-point response to the comments.

**Reviewer #1**

**1 General Comments**

*The authors present a study on the Bromine-MION CIMS, an impressive new tool to selectively and extremely sensitively measure several relevant iodine species and sulfuric acid at the CLOUD chamber with clearly the same potential for atmospheric measurements. The combination of an extremely clean chemical ionization procedure with a very high-resolution time-of-flight mass spectrometer renders unprecedented selectivity and obviously sensitivity which is shown by means of instrument intercomparisons and employing accurate absolute calibration techniques as well as a rough method based on intercomparisons of measured and ab-initio calculated cluster-ion formation and dissociation enthalpies. Overall this is a very relevant study fitting perfectly into the scope of AMT.*

*However, the way the involved experiments are presented and discussed needs considerable revisions in order to enhance the usefulness to the CIMS community and also be comprehensible for non-specialist readers. I give my general concerns in the following paragraphs and many detailed comments and proposals in the next section.*

*In general the manuscript is written in a way hard to follow for non-specialists in the CIMS technique. Just as one example the widely used ion signal ratios should be briefly rationalized in terms of ionization reaction rates before being discussed. Although the paper refers to the relevant instrument papers basic procedures and derived quantities etc. used in 'every day live' by the authors must be briefly explained. The reader cannot be expected to look up 10 other papers in order to properly follow the basic course of the manuscript.*

Reply: Ion-molecule reactions are typically second order reactions, fluctuations in reagent ion concentration may cause small variations in formation rate of ion-molecule clusters, affecting the analyte signal. We have now added a sentence to explain this (Lines 318-324 of the revised manuscript):

"Chemical ionization relies on an ion-molecule reaction to transfer charge from a reagent ion to an analyte, forming either a product ion or a charged cluster between the analyte and the reagent ion with a rate coefficient $k_{IM}$. This occurs in an ion-molecule reactor, with a fixed flow rate and thus reaction time, $dt$, and ideally under pseudo first-order conditions where a small fraction of the analyte is ionized and the reagent ion concentration ([Ion]) remains constant. Under these (linear) conditions the fraction of analyte that is ionized is $k_{IM} \times$ [Ion] $\times dt$. However, the primary ion source strength can vary with time, and so we normalize the analyte signal by reagent ion signal to account for those small variations in analyte signal."

*In places sloppy and unclear formulations are used, e.g. 'uncertainty' instead of clearly stating if accuracy or precision are referred to and if 1 or 2 standard deviations are given.*

Reply: We thank the reviewer to point out this vague language. We have extensively replaced "uncertainty" with "1-sigma accuracy".

*Also details of the experiments need to be much better defined where the relevant parameters have clear influence on the results presented. As an example critical settings of electrical fields employed in the first transfer stages which have strong impact on the relative signals of ion clusters and bare ions in the mass spectra are not even mentioned. Reproducibility of the results presented is not given if the reader has to try to isolate such relevant information from the papers cited and can't be sure that they have not been changed. As another example it is not even mentioned which type of mass analyzer is used for the MION-CIMS or also the FIGAERO and what the relevant mass resolutions are. Therefore it needs to be clearly stated in the methodology section 2 in which relevant operational conditions the instrument(s) have been operated. Pressures and flows employed in the IMR and 1st transfer stage and respective reaction times (as well as the electric fields applied in the ion transfer) should be given in the experimental section and not implicitly towards the end of section 3! This is absolutely necessary for sake of reproducibility. A table may do the job here.*

Reply: We have listed all relevant instrument specifications and operational conditions in Table S1 according to the reviewer's suggestion, and included a statement on instrument tuning (Lines 158-163 of the revised manuscript):

"We optimize the adduct-ion signals in both the Br-MION-CIMS and Br-FIGAERO-CIMS by tuning the electric field strengths in the first two low-pressure stages of the mass spectrometer as weak as possible to minimize collision induced cluster fragmentation while maintaining sufficient ion transmission. Optimization is achieved by maximizing the ratio of $I_2Br^-/Br^-$ at a constant $I_2$ concentration. We list relevant instrument specifications and operational conditions in Table S1. It should be noted that these values are specific to our instruments, thus can vary according to instrument parameters and may not be applicable to other instruments."

*In places purely qualitative statements in the discussion should be quantified by giving the relevant parameters such as for relative reaction rates etc. (see in the specific comments).*

*For the I2 and Cl2 permeation devices which have been nicely qualified I am wondering why no parallel gravimetric control has been done as this is a very simple and reliable way of calibrating devices with such compounds and high permeation rates approaching the mg/day regime.*

Reply: We agree that the gravimetric analysis will be another way of calibrating the permeation devices. However, we did not have a balance in our laboratory with precision that is able to measure the weight changes of the permeation device, so we were unable to setup the parallel gravimetric control system during our calibration experiments. Therefore, we decided to use chemical analysis, which is a widely used method, to quantify the permeation devices.

*The operations around the permeation devices have been worked out in quite some detail although these procedures are not new but have also been described in earlier publications which are cited. I don't criticize that detail, however, there is a strong misbalance to other experimental details especially since this part has been placed into the results and discussion section 3. Here certainly some restructuring is necessary.*

Reply: We have moved the experimental details to the methodology part in "section 2", and restructured the "section 3.3.2" of the revised main text according to the reviewer's suggestion.

*Two independent approaches for the I2 calibration have been employed and presented. Both look very good but differ by more than a factor two from the CE-DOAS intercomparison to the permeation tube approach most obvious in Table 3. This is not clearly mentioned nor discussed in the manuscript. On the contrary the summary states that 'different methods result in consistent calibration constants'!? Since both methods used are in principle absolute techniques and individual errors seem much lower there must be some conclusion on this discrepancy seemingly outside the combined error bars. Has the DOAS system been calibrated directly employing the permeation device in order to resolve this? This point clearly needs revision and clarification!*

Reply: The reviewer is correct that we describe a difference of a factor of approximately 2.3 for the calibrations of $I_2$ by CE-DOAS and permeation tube calibrations. However, these two calibrations were carried out with different instruments and slightly different settings, at different times. Due to limitations of available equipment, we were unable to compare the two methods simultaneously. The manuscript was insufficiently clear in this regard. The main take-home message is that in-situ calibration is critical. In this context we regard the agreement within a factor of 2.3 as reasonable; these tuning differences require in-situ calibration but indicate that the instruments are consistent, but the final factor of 2 requires careful calibration. In our case, one of the calibrations (by CE-DOAS) was done onsite at CLOUD while the off-site calibration (by a permeation tube) was carried out in a laboratory at the University of Helsinki. The reviewer is also right that it is misleading to state that the results are consistent. Our initial intention was stating that calibration factors were not inconsistent, since a factor of two difference in two independent calibrations is common in the characterization of the chemical ionization method we use. Three changes are made in the revised version:

1. We have deleted the statement about the consistency in the calibration coefficients;
2. We added discussions about the difference in the inter-method (in-situ) and offline calibrations and highlighted the importance of carrying out calibration experiments at individual experiments;
3. We added a table which lists the key parameters of the Br-MION-CIMS in the inter-method and offline experiments.

*An approach to compare the performance of the MION and FIGAERO instruments especially with respect to surface effects is presented in section 3.4.2 on rise and decay time constants. The tentative interpretation presented seems reasonable, however my feeling here is that the fitting analysis does not give any significant additional information on the signal evolution than obvious already from the pure normalized signals. MION and FIGAERO agree well for HIO3 and HOI. Therefore the evolutions probably indicate wall uptake and desorption processes possibly occurring somewhere from the chamber into the IMRs since else certainly much slower decays would be expected. The fitted evolutions just underline the instrument deviations which seem really significant only for IO with the obvious elevated background signals on the FIGAERO. This might also be due to some residual outgassing of the FIGAERO filters after the aerosol measuring phases which just ended before start and end of the photolysis. Another reason for*

*the 'permanent' IO background in the FIGAERO might also be some unrecognized isobaric contaminant on the lower resolution (if I'm not mistaken) FIGAERO analyzer. However, in the current state of analysis I don't see any additional information provided by this analysis technique.*

Reply: The reviewer's point is well-taken. In order to condense the discussion, we have removed Figure 6, and combined the "Signal trend and detection limit" and "Rise and decay time constants" sections. We agree with the reviewer that decay of $HIO_3$ is driven by wall uptake. But for IO, it's less likely that the IO background comes from residual outgassing of the FIGAERO filter, because we did not observe much IO signal during particle-phase analysis (not shown in the manuscript); it is also less likely that it comes from an unrecognized isobaric contaminant, because FIGAERO uses the same type of mass analyzer as the MION (mass resolution ~10000, we have listed the instrument specifications in Table S1, as suggested by the reviewer). We argue that at least the decay rate analysis would be useful for physical chemists who want to simulate the chamber experiments and investigate kinetics of the iodine oxidation reactions. Thus, we have preserved the key results, but condensed the lengthy discussion.

**2 Specific Comments**

*l.116: Here an outline of a typical NPF experiment on the CLOUD chamber should be briefly presented with the operational details of the experiment, like flushing, I2 addition and control, etc as a function of experiment time. See general comments.*

Reply: Following the reviewer's suggestion, we now have added more operational details of the experiment in "section 2.1":

"The dry air supply for the chamber is provided by cryogenic nitrogen (Messer, 99.999 %) and cryogenic oxygen (Messer, 99.999 %) mixed at the atmospheric ratio of 79:21. Ultrapure water vapor, ozone and other trace gases can be precisely added to attain desired mixing ratios at different levels. The total injection rate of the humidified air is fixed at 330 standard liters per minute (slpm) to compensate for the sampling consumption of various instruments. Molecular iodine ($I_2$) is injected into the chamber by passing a flow of cryogenic nitrogen through a crystalline iodine ($I_2$, Sigma-Aldrich, 99.999 % purity) reservoir, which is temperature-controlled at 10 ℃, to achieve levels of 0.4 to 168 pptv in the chamber. The sulfinert-coated injection lines are temperature stabilized to minimize line conditioning effects. High intensity green light emitting diodes (LEDs) are used to photolyze molecular iodine into iodine atoms and initiate the subsequent oxidation reactions in the presence of ozone and water vapor. The 48 green LEDs (light sabre 4, LS4) are mounted in pairs (one upward facing, one downward facing) on a copper cooling bar within a long quartz jacket that protrudes into the chamber in the mid plane. The maximum total optical power output is 153 W, centered on 528 nm. Actinic fluxes are regulated by controlling the number of LEDs used and the set point of individual LEDs. Light fluxes are monitored by two photodiodes and a spectrometer."

We would like also to refer to "section 3.1", where we had described the sequence of a typical iodine oxidation experiment.

*l.130: Details of the experiments need to be much better defined, see general comments.*

Reply: This comment has been addressed in the general comments by adding the Table S1 and discussions in the "section 2.1" and "section 3.3".

*l.166: Accuracy is a defined quantity and I'm not aware of 'systematic accuracy'. Is 20% accuracy or rather precision meant here? For all statements on accuracy or precision it must be clearly mentioned which quantities are given, 1 or 2 standard deviations?*

Reply: We thank the reviewer to point out this vague language. In this specific case, 'accuracy' is meant. We have reworded the sentence (Lines 168-169 of the revised manuscript):

"The overall accuracy for the $I_2$ time series is estimated to be 20 %, never better than the detection limit (3-sigma precision), resulting from the uncertainty in cross sections and the stability of the baseline."

*l.171: Isn't 'conformational sampling' the more common term?*

Reply: Corrected.

*l.179: The experimental details on the qualification of the I2 and Cl2 permeation devices and HOI calibrator operation from section 3 should be moved to here. Clearly these are methodological details. See general comments.*

We have moved the experimental details to the methodology part in "section 2" and restructured the "section 3.3.2" of the revised main text according to the reviewer's suggestion.

*l.187: ... high mass resolution ... Here the mass analyser type and important specs should be detailed!*

Reply: We have listed important specifications in Table S1.

*l.222: Does the RH range encompass all relevant experiments carried out? Lee et al., 2014, show that for I- ionization the major RH sensitivity of the calibration, especially for small organic acids, is found at quite low RH.*

Reply: As the initial motivation of study was focused on measuring gaseous iodine species under conditions relevant to the marine boundary layer, calibration experiments were carried out at RH between 20 % and 40 % at 25 °C, and between 40 % and 80 % at -10 °C. Therefore, we can only report that this technique is independent of RH under these conditions. We did not test the RH effect at very low RH, as it was beyond the scope of our initial motivation; but it would certainly be interesting to perform more calibration experiments covering the full RH range in the future. If we were to speculate, there could be three reasons why Lee et al., 2014 found a certain RH dependence, while we did not:

1.  The inlet used in Lee et al., 2014 was a low-pressure ion-molecule reaction (IMR) chamber, operating at 90 mbar, while the MION inlet is an atmospheric pressure inlet. Different water concentrations and ion-molecule reaction timescales in the IMRs could result in different RH dependences.

2.  The RH effect differs for different species, as also discussed in Lee et al., 2014. We show that the Br-MION-CIMS has a negligible RH dependence for $I_2$, HOI, ICl and IBr; however, it can have a different dependence toward other species, e.g. organic acids.

3. The bromide chemical ionization may behave differently from the iodide chemical ionization in terms of detectability and RH dependence (less likely).

Currently, it is not fully understood which one(s) of the above-mentioned reasons are responsible for the observed difference. Therefore, we have revised our discussion about the RH dependence (section 3.2) to restrict application of our conclusion.

*l.228-232: It may be worthwhile to although show the respective plot for the HOI normalization.*

Reply: We agree with the reviewer that this effect warrants explicit mentioning. The effect of humidity is already inherently included in Figure 4, where we show the results for HOI calibration experiments. During the calibration, we varied the water content in the calibrator to vary OH concentrations. The good linear correlation between HOI concentration and normalized HOI signal indicate water vapor concentration did not affect the detection. We have now linked this statement in the text to the Figure 4.

*l.246: I propose to not present the rather lengthy equation in the text flow but as a separate equation.*

Reply: Done.

*l.246: Is there a physical motivation for the inclusion of the quadratic fit? If there is any relevance, apart from the good agreement with the linear relation, it should be clearly discussed. Else it can be neglected.*

Reply: We concur with the reviewer that the (linear) sensitivity is of primary interest for the scope of the article. . We do not have a conclusive explanation for the observed deviation from the linear sensitivity and would prefer not to speculate. Following the suggestion, we have removed the quadratic fit from Figure 3, and modified the text accordingly (Lines 356-365 of the revised manuscript):

"As shown in Fig. 3, we use the accurate $I_2$ time series measured with the CE-DOAS to calibrate normalized $I_2$ signals in the Br-MION-CIMS. The $I_2$ concentrations used for the calibration span approximately 2 orders of magnitude, reaching up to $4.6 \times 10^{10}$ molec cm$^{-3}$. A linear fit, limited to $I_2$ concentrations smaller than $10^{10}$ molec cm$^{-3}$, establishes the calibration factor as follows:

$$[I_2] = 2.7 \times 10^{10} \text{ molec cm}^{-3} \times I_2 \cdot {}^{79}Br^- / ({}^{79}Br^- + H_2O \cdot {}^{79}Br^-)$$

For this range of concentrations, which are typically encountered in the atmosphere, the calibrated Br-MION-CIMS time series agrees within error with the CE-DOAS measurement (1-sigma accuracy 20 %, detection limit 25 pptv for 1 min data). Deviations between both time series are generally smaller than 10 % (25 and 75 percentile 0.88 and 1.03, respectively). These small differences may be explained by incomplete homogeneity of iodine concentrations in the chamber and the different sampling positions of CE-DOAS and Br-MION-CIMS."

*l.255: For this as well as the Cl2 device all wetted materials used for the calibration gas generation should be detailed. Especially for the low flow rates used for flushing the permeation devices even compounds like the elemental halogens could be reduced due to surface effects.*

Reply: We thank the reviewer for bringing up this concern. However, we are convinced that the surface effects (if any) are less likely to affect our calibrations for the following reasons. First, we use dry $N_2$ as the carrier gas to flow over the permeation devices. Second, once the gas flow with $I_2$ or $Cl_2$ leaves the permeation tubes, it immediately passes into the absorption vessel; there is no surface exposed to moisture during the transmission. Third, even if there are some surface effects, we expect them to be cancelled out as we use the exact same permeation devices to calibrate the Br-MION-CIMS.

*l.280: Quantification would clearly help with this statement (e.g. ... agreement within XX% with ...).*
Reply: We have specified the agreement between two methods (Lines 220-221 of the revised manuscript):
"The ICP-MS results were in good agreement (within 20 % discrepancies) with those from the UV/Vis spectrophotometry."

*l.290: The calibration constant reported for I2 from the CE-DOAS intercomparison was 2.7E − 10 molec/cm−3, i.e. more than 60% lower as mentioned earlier. However, were does the 'overall uncertainty' (is this accuracy?) of 45% come from while the permeation rate is defined to better than 5%?*
Reply: Thanks for pointing this out. We need to clarify that the instruments setup used for these two experiments is slightly different, such as sample flow, sample line geometry, etc. This would lead to differences in calibration coefficients. The overall uncertainty here refers to the summed uncertainties during the $I_2$ calibration using the permeation device, including the fluctuations in the permeation temperature, accuracy of the ICP-MS during the chemical analysis, signal deviations from 5 calibration experiments, and statistical error of the fittings; the standard deviation of the permeation rate at a certain temperature is just a subset.

*l.336: Give the relative reaction rate instead of ' ... relatively slow ...'*
Reply: We have revised the statement in the text as the following (Lines 257-261 of the revised manuscript):
"We only included the formation pathway of $I_2$ + OH to HOI in the model for simplicity; the other pathway of IO + $HO_2$ was considered minor as its reaction rate is about an order of magnitude slower than that of $I_2$ + OH. Furthermore, IO is likely at negligible concentration in the calibration device due to the absence of $O_3$ for IO formation via I radical and $O_3$ to take place forms at a relatively slow rate via the reaction of I radical and $O_3$."

*l.351: No statements on the accuracy of the H2SO4 measurements for the pre-calibrated nitrate-CIMS nor the Br-MION are given in this section. Just the caption of Fig.4 mentions a 'systematic scale uncertainty'. This is very annoying and makes me wonder if this has been left out on purpose ...*
Reply: We apologize for not having had the manuscript clear in this point. We did not leave out any information on purpose. The accuracy estimation of $H_2SO_4$ calibration using a $H_2SO_4$ calibrator is described in details in Fig S5 in Stolzenburg et al., 2020. The systematic 3-sigma accuracy for $H_2SO_4$ measurement using exactly the same nitrate-CIMS and exactly the same calibration method is +50/-33 %. When two mass spectrometers are compared, there are additional two minor sources of uncertainty: the statistical error of the signal correlation analysis (1.1 %) and the error

of small variations in analyte signal caused by reagent ion fluctuation (1.7 %). Eventually, the propagated error is +50/-33 %. We have added a sentence to specify this (Lines 377-378 of the revised manuscript):

"The systematic 3-sigma accuracy is +50/-33 % for $H_2SO_4$ calibration using a nitrate-CIMS; detailed accuracy estimation has been discussed previously (Stolzenburg et al., 2020)."

*l.348: I propose to not present the rather lengthy equation in the text flow but as a separate equation.*
Reply: Done.

*l.374: The explanation is hard to comprehend for the non-specialist reader and rather leads to confusion. This should be improved upon e.g. by going step by step from the dissociation enthalpies via reaction rates etc.*
Reply: We have revised the text to make it more straightforward (Lines 425-433 of the revised manuscript):

"where the X-H is the hydrogen bond donor. An analyte should be detected at the maximum sensitivity when the dissociation enthalpy for the first pathway is either a) much higher than the critical enthalpy (dissociation of X-H·$Br^-$ to X-H and $Br^-$ does not occur), or b) lower than the critical enthalpy, but much higher than that of the second pathway (dissociation of X-H·$Br^-$ to X-H and $Br^-$ may occur, but dissociation to $X^-$ and $HBr$ is the dominant pathway). Whether the enthalpy for the second pathway is higher than the critical enthalpy does not directly affect the sensitivity, as both X-H·$Br^-$ and $X^-$ can be measured and counted. The sensitivity toward X-H would be reduced only when the first reversion pathway occurs to a non-negligible extent. Taking $H_2SO_4$ as an example, the dissociation enthalpies for the first and second pathways are 41.1 and 27.9 kcal $mol^{-1}$, respectively. If some of the $H_2SO_4$·$Br^-$ dissociate, they preferably become $HSO_4^-$ and are detectable by the Br-CIMS. Thus, $H_2SO_4$ can be detected at the maximum sensitivity."

*l.393ff: Even though the derived calibration constants (and later detection limits) of the species not calibrated directly are quite crude they should be given in a table rather than in the text flow. I propose to include them into Table 3 as a third category c: derived from dissoc. enthalpies (or similar). Table 3 could be extended by another column for the det. limits for all species as well.*
Reply: We thank the reviewer for this suggestion and have now included the calibration coefficients and detection limits in Table 3.

*l.421: I find the use of 'NPF event' rather annoying since no particle data are shown or discussed, I propose to just refer to the 'photolysis interval' or similar.*
Reply: We have replaced "new-particle formation experiment(s)" with "iodine oxidation experiment(s)" extensively.

*l.429: Is there a continuous addition of I2 so to keep it constant over the photolysis causing I2 to increase when the lamps are off?*
Reply: Yes, the $I_2$ was continuously added during this experiment. We have added in line 510 "while its injection continues" to clarify.

*Fig.1: I really do like the figure, however, the caption will have to be extended considerably in order to properly inform and guide the reader! First there should be 2 defined panels (upper and lower, a) and b), ...). Mass defect could be defined in the text so not to extend the caption. More clear identification of the experiment (date, ...) may not be absolutely necessary here but still enhance transparency. Upper panel: symbol area is proportional to the logarithm of the signal AREA which has a unit of counts/s. However, then the y-axes in the lower panel can not have the same unit! Please, align these details!*

*Lower panel: The explanations given in the upper left spectrum must be detailed. Although obvious it should be indicated that colors in the panels do match. It might be sensible to use the same x-scaling for all spectra. What is the averaging time of the spectra shown?*

*Fig.2: The time series plot shows peak area (not intensity) vs. time.*

Reply: We have included the experiment date in the figure caption, defined the upper and lower panels as Fig 1(a) and (b), respectively, clarified the legend of Fig 1(b), and indicated that species in both panels are color-coded in the same style. The averaging time of the spectra is one hour.

For the unit of spectra and the time series in Fig 1 and Fig 2, the reviewer has a good point, but we are inclined to follow the unit convention for CIMS (e.g. Fig 1 in Lee et al. 2014), and continue using units of counts per second.

*Fig.3: A different and zoomed color scale has to be selected here in order to better show the range of RH varied from just 40-80%. Is there an explanation for the slight hockey stick shape visible at high RH and low I2 concentrations? Is this a consequence of the DOAS det. limit?*

Reply: We agree with the reviewer that a simplified color scheme is sufficient, clearer, and helps to more efficiently illustrate the influence of humidity on the measurements. In fact, the distribution of humidity justifies the use of two discrete colors for "dry" and "wet" conditions in the revised figure. Following the main text comment we eliminated the quadratic fit. The non-linearity may be explained by inhomogeneity of $I_2$ in the chamber. As this is already discussed in the main text, we do not repeat it in the figure caption. To reflect the change of the figure, we slightly modified the caption (Lines 340-345 of the revised manuscript):

"Signal normalization methods for the Br-MION-CIMS. Normalized $I_2 \cdot Br^-$ signal intensity for variable $I_2$ concentrations, color coded by relative humidity (orange: 35-45 % relative humidity, red: 70-80 % relative humidity). The charger ions in the ion source of Br-MION-CIMS are $Br^-$ and $H_2O \cdot Br^-$ (both $^{79}Br$ and $^{81}Br$). Their abundance depends both on the instrument tuning and the absolute humidity of the sampled flow. The normalization of the $I_2 \cdot Br^-$ signal by only $Br^-$ (a) or $H_2O \cdot Br^-$ (b) does not compensate for the humidity effect. Using the sum of $Br^-$ and $H_2O \cdot Br^-$ (c) for normalization yields a tight correlation to the true $I_2$ as measured by CE-DOAS, independent of the humidity. The black dashed line indicates the fitted linear calibration."

*Fig.4: The x-axis titles should give the detected ion clusters. The analyte species may be given in the caption.*

Reply: We respectfully disagree with the reviewer, because the data points here are not raw signal for ion clusters, they are normalization signal as we discuss in "section 3.2".

*Fig.1: abc*

Reply: Done.

**4 Minor comments, typos, etc.**

*l.70: I recommend to use 'absorption features' instead of cross sections.*

Reply: Corrected.

*l.83: The bromide ion ...*

Reply: Corrected.

*l.114: ... the atmospheric oxygen ratio of 0.21 ... (there aren't 79% of N2!)*

Reply: Corrected.

*l.186: ... bromide ionS ...*

Reply: Corrected.

*l.247: ... the whole campaign ... THE two curves ...*

Reply: Corrected.

*l.327: ... hydroxyl radicalS ...*

Reply: Corrected.

*l.350: This is a repetition from the equation given before ...*

Reply: We have reworded the sentence (Lines 376-377 of the revised document).

*l.414: ... of THE bromide ...*

Reply: Corrected.

*l.423: ... and TO low levels of I2... (The sentence may otherwise suggests that I2 had not yet been added to the chamber at this point.)*

Reply: Corrected.

*l.446: ... of THE NPF event ...*

Reply: Corrected.

*l.739, 743: ... with bromide ionS.*

Reply: Corrected.

**Table S1: Instrument specifications for Br-MION-CIMS and Br-FIGAERO-CIMS.** The values are specific to our instruments, thus can vary according to instrument parameters.

| Instrument specifications | Br-MION-CIMS (lab) | Br-MION-CIMS (CLOUD) | Br-FIGAERO-CIMS |
|---|---|---|---|
| Total sample flow (slpm) | 20 | 32 | 18 |
| IMR pressure (mbar) | 1000 | 1000 | 150 |
| IMR residence time (ms) | 30 | 20 | 200 |
| SSQ pressure (mbar) | 1.9 | 2.2 | 2.0 |
| BSQ pressure (mbar) | 0.011 | 0.012 | 0.011 |
| [a]MION accelerator (V) | -2750 | -2800 | n/a |
| [b]MION deflector (V) | -210 | -290 | n/a |
| Nozzle (V) | 11.4 | -1.4 | 0.05 |
| SSQ EP (V) | 13.79 | -5.13 | 0.10 |
| SSQ front (V) | 20.92 | 12.45 | 0.20 |
| SSQ back (V) | -16.94 | 8.22 | -0.30 |
| Lense skimmer (V) | -16.17 | -6.16 | -0.19 |
| Skimmer (V) | 0.59 | 0.63 | 4.00 |
| BSQ front (V) | 4.78 | 1.76 | 5.89 |
| BSQ back (V) | 4.93 | 2.76 | 5.90 |
| Skimmer 2 (V) | 5.77 | 5.69 | 5.92 |
| Reference (bias) (V) | 124.94 | 105.87 | 121.99 |
| Ion-lense (V) | 34.076 | 31.21 | 39.78 |
| Deflector flange (V) | 53.79 | 75.91 | 88.31 |
| Deflector (V) | 61.77 | 83.82 | 95.08 |
| Mass analyzer | microchannel plate detector | microchannel plate detector | microchannel plate detector |
| Mass resolution | ~ 10000 | ~ 10000 | ~ 10000 |

[a] The voltage to accelerate the reagent ions in the ion source (Rissanen et al. 2019).

[b] The voltage to deflect the reagent ions to the sample flow (Rissanen et al. 2019).

**References**

Lee, B. H., Lopez-Hilfiker, F. D., Mohr, C., Kurten, T., Worsnop, D. R. and Thornton, J. A.: An iodide-adduct high-resolution time-of-flight chemical-ionization mass spectrometer: application to atmospheric inorganic and organic compounds, Env. Sci Technol, 48(11), 6309–6317, doi:10.1021/es500362a, 2014.

Passananti, M., Zapadinsky, E., Zanca, T., Kangasluoma, J., Myllys, N., Rissanen, M. P., Kurtén, T., Ehn, M., Attoui, M. and Vehkamäki, H.: How well can we predict cluster fragmentation inside a mass spectrometer?, Chem. Commun., doi:10.1039/c9cc02896j, 2019.

Stolzenburg, D., Simon, M., Ranjithkumar, A., Kürten, A., Lehtipalo, K., Gordon, H., Ehrhart, S., Finkenzeller, H., Pichelstorfer, L., Nieminen, T. and others: Enhanced growth rate of atmospheric particles from sulfuric acid, Atmos. Chem. Phys., 20(12), 7359–7372, 2020.

---

## Author Comment (AC2) · 8 Apr 2021

**RE: A point-by-point response to referee comments**

We thank the reviewer for dedicating time for a diligent review of the manuscript, and for providing valuable comments and detailed suggestions for improvements. We have carefully modified the manuscript to improve the legibility, to clarify critical instrument parameters, and to specify the potential limitation of this method in laboratory and field measurements. Below we provide a point-by-point response to the comments.

**Reviewer #2**

*This is a well written and organized account of the application of a bromide cims instrument for the detection of various iodine containing species as well as sulfuric acid. The authors present nice account with much of the pertinent information related to their experiments. However, I find that there is much to be desired with respect to the errors associated with the methods used here and the various approximation used through-out the analysis. This is significant as the authors are presenting a very optimistic account of the ability of this method to detect extremely low levels of these species in a real-world marine environment. If in fact the authors can provide more details and support their conclusions through the additional information this is a truly remarkable performance and will be of wide interest to the field of atmospheric mass spectrometry. It is my opinion that more detail must be provided to support the publication of this manuscript in AMT and therefore recommend the following minor revisions to the manuscript.*

*I am having a difficult time understanding the true errors associated with your calibration factors presented throughout the manuscript. In some cases, you are calibrating using external calibration sources and in others you are calibrating using a secondary instrument for comparison. One issue with this whole description is the lack of information provided for exactly how these were performed. What I mean by this is how were the calibrations sources samples, through an inlet or directly into the IMR? Or when two instruments were compared were different inlets used between the two instruments. This is of particular importance when extrapolating the cluster binding energy approximated sensitivities to your instrumentation. I ask all of this because the actual true sensitivity to a given species is a combination of the ionization sensitivity as well as an instrument function which describes any inlet losses or differences in ionization potential between various instruments.*

Reply: For the offline calibrations, the calibration sources were directly sampled using the MION inlet after dilution; for the inter-method calibrations, the CLOUD chamber acted as the sources, and the chamber flow was directly sampled using the same MION inlet. In order to make the description clearer, we have now listed the instrument specifications and the operational conditions in Table S1.

*The main point of your work is that this method can detect these species with the sensitivity and detection limits sufficient for real world marine environments, so this really matters here. How well do you actually know your sensitivities and detection limits? Are those an upper limits because they represent sensitivities without an inlet present? You have made many assumptions related to the calibration of your instrumentation and that is certainly not reflected in any robust error analysis that is provided to the reader.*

Reply: The sensitivity toward a specific compound depends on two factors: 1) whether a compound can be charged by the reagent ion to form a stable ion (cluster); 2) whether the formed ion (cluster) can be guided through the relatively low-pressure regions (by electric fields) and be detected by the APi-TOF. The Br-CIMS is able to detect a compound only when both of these two factors are satisfied. If the compound clusters strongly with the $Br^-$ ion, and a minimum delustering occurs in the charging processes and in the low-pressure regions, the compound can be detected at the kinetic limit (e.g., $I_2$ and $H_2SO_4$), thus at a high sensitivity.

The detection limit mainly depends on how well we are able to separate the compound signal from background noise (signal to noise ratio). Such signal identification can be compromised by two factors: 1) an undesired peak that covers the compound peak; 2) background signals produced by the electronics, which appear as low-level signals even when there is no compound appearing at that mass to charge ratio ($m/z$). The first factor is largely overcome by the high mass resolution of our instrument (~10000, Table S1). Thus, the major factor that determines the detection limit is the background noise. Therefore, we define the detection limit in this study by the mean value + three times the standard variation during a two-hour time period in which we know the analyte compound is absent. If a compound peak appears at the concentration as defined by the detection limit, we can easily separate it from the background noise.

All in all, with the instrument calibrations either by offline methods or the inter-instrument methods, we quantify the sensitivity which accounts for the charging processes in the MION and FIGAERO inlets (flow rate, pressure, ion-molecule collision, etc.) and ion transmission in the mass spectrometer. With the analysis on the instrument background, we define conservatively the detection limit toward a compound.

The defined detection limits are all conservative values and rather an upper limit, and often even better detection limits are realized by the instruments (i.e., they are able to measure even lower concentrations). All the detections limits are defined when the inlets are connected to the instrument, thus the detection limits contain the information from the calibration and instrument background.

*On the topic of using cluster binding energies for the approximation of sensitivities, the authors suggest that because that sort of analysis has worked for iodide than it naturally works for the Br- ion chemistry, however, there is no evidence that this approach is true. Again this method is also complicated by the differing inlet response functions of the various species and would really only yield an upper limit to the ionization sensitivity not the true detection capabilities of a sampling system. For example, a binding energy does not take into account that sulfuric acid has a different transmission factor than IO which would also vary with sampled ambient humidity on the inlet surfaces.*

Reply: Passananti et al. (2019) studied the collision induced fragmentation in the same instrument. It has been concluded that the sensitivity of our instrument largely depends on the cluster formation enthalpies of the analyte-reagent ion cluster and the tuning of a specific instrument. With a specific tuning, a strongly binding cluster has a much lower chance to fragment compared to a less strongly bound cluster. This conclusion applies not only to $I^-$ ion chemistry but also to, e.g., $NO_3^-$ and $Br^-$ ion chemistry.

The reviewer is correct that the binding energy discussion does not consider the different diffusivity of different species. In the revised version, the only calibration factor we transferred is the one for $HIO_3$. We calculated that the inlet line loss for $H_2SO_4$ and $HIO_3$ are 33 % and 36 %, respectively. The calculation was based on a hard sphere

assumption (for the calculation of the diameters of the two molecules), an inlet flow rate of 32 slpm and an inlet length of 1.53 m. As such difference is marginal, we believe that the calibration factor of $H_2SO_4$ is applicable to $HIO_3$. However, to accommodate the reviewer's suggestion, we have deleted our estimation on the exact calibration factor of IO and OIO. And we added a note in the section 3.3.3 that the difference in the diffusivity needs to be considered in transferring the calibration factor.

*I think the calibration through instrument comparison method was used in this manuscript to transfer the MION calibrations to the FIGAERO, perhaps I incorrectly understood something. If that is the case, this issue of instrument functions becomes even more of a hindrance to the direct comparison of ambient observations because the two instruments operate at very different pressures with significantly different ionization schemes. These types of issues clearly need to be stated and incorporated into the stated performance of the instruments, preferably through a more robust error analysis. Although that may not be possible as the error in approximated calibration factors may not be available.*

Reply: The reviewer is correct that the Br-MION-CIMS and the Br-FIGAERO-CIMS use different inlets and operate at different pressures. These differences would indeed lead to different calibration coefficients in the two instruments for the same analyte. However, it is not the calibration coefficients that we transfer from one to another, we transfer the calibrated absolute concentrations, and only for analytes whose signals from the two instruments are linearly correlated, such as $HIO_3$ and HOI. Thus, the major error really comes from the calibration sources, e.g. +50/-33 % for $HIO_3$ and 55 % for HOI.

When two mass spectrometers are compared, there are two minor errors: the statistical error of the signal correlation analysis (e.g. 3.1 % for $HIO_3$ and 3.2 % for HOI) and the error of small variations in analyte signal caused by reagent ion fluctuation (2.9 % for all analytes). Eventually, the propagated errors are +50/-33 % for $HIO_3$ and 55 % for HOI. We do agree with the reviewer that a more robust error analysis is important. We have added a statement in Lines 517-519 of the revised manuscript:

"Note that for both HOI and $HIO_3$ the uncertainties introduced from the correlational analysis are negligible compared to the limited accuracy of the calibration sources (55 % for HOI and +50/-33 % for $HIO_3$)."

Besides, we have listed the key specifications of the two instruments in Table S1 to clarify the differences.

*I would like to know more details on the what the background conditions were during the periods for which the detection limits were calculated. It is well known that when a CIMS instrument samples clean air the background values are continually reduced. In most real world applications conditions are far from clean and as such I would expect a relatively larger H2SO4 background than what the authors are stating. I am curious if this discrepancy comes from the fact that this instrument is sampling from what is perhaps the cleanest atmosphere available on earth which allows for unrealistic instrument backgrounds. Yes, that would be the functional instrument detection limits for sampling in the CLOUD chamber, however the results would not be directly translatable to a real-world environment where instrument background are likely significantly higher. If in fact the detection limits are determined from periods of chamber measurements, or even if the instrument is only sampling from the chamber and not room air then the*

*authors should admit that this is really a best-case scenario for the stated detection limits. This would mean that real-world applications of this method in the marine atmosphere are unlikely to achieve these results. Unless of course the authors can support the idea that these instrument backgrounds are routinely achievable on a standard field deployable CIMS instrument. If this is the case data should be shown describing the various background and operating conditions. Also in general, during the chamber experiments how were zeros performed?*

Reply: The reviewer's point is well taken. However, we can justify that these instrument backgrounds are routinely achievable on a standard field deployable CIMS instrument. Other than the CLOUD measurements, in the summer of 2018, we have deployed the same bromide-CIMS for an intensive field measurement at Mace Head, Ireland (for details please refer to our previous publication: Tham et al., 2021). During the campaign, we conducted 4 separate background measurements, by flooding the inlet with excess nitrogen flow (99.999 % of purity) for at least 10 min during the daytime low tide event (typically has significant $H_2SO_4$ levels, up to 4 pptv). The results show the absence of the iodine species of interest when compared to the ambient spectra, and the calculated detection limit (3σ) for $H_2SO_4$, $HIO_3$, $I_2$ and HOI are $7.4\times10^5$, $2.2\times10^6$, $1.6\times10^6$, and $2.1\times10^6$, molec $cm^{-3}$, respectively.

Background levels and detection limits are quite comparable during the CLOUD experiments. We achieve the background conditions by terminating photochemistry and subsequent formation of oxidized iodine species/$H_2SO_4$, while keeping the 330 slpm humidified air constantly flushing the chamber system. In the meantime, we turn up the fan speed to increase the vapor wall-loss rates.

Low background level for $HIO_3$ and $H_2SO_4$ can be achieved because they have very low volatility and are irreversibly lost to surfaces, and even $10^7$ to $10^8$ $cm^{-3}$ $HIO_3$ or $H_2SO_4$ (ambient upper limit) would not saturate the sample line walls. For more volatile species such as HOI, partitioning between the gas phase and the wall may increase the background level (although it was not observed in our experiments), which is dominated by vapor concentrations (gas-phase activities). However, the CLOUD experiments run at atmospherically relevant vapor concentrations and conditions, and it is very likely these results are translatable.

*The pressure at which these instrument are being run at leads me ask if the authors have identified any issues with the combination of Br- and O3 as it relates to secondary ion chemistry. It is fairly well known that the addition of O3 to an I- CIMS will result in the production of IO-, IO2-, and IO3- which can act as a secondary reagent ion even at low abundances. This presents challenges to interpretation of oxygen rich ions where (HIO2)Br- can actually be the result of a cluster of HOI with BrO-. The pressures used in this study are sufficiently high in an iodide CIMS to result in these reactions at this ozone level and I would encourage the authors to discuss this potential.*

Reply: We thank the reviewer for this insightful question. We checked a typical experiment carried out at -10 ℃ and did not find the $BrO_3^-$ in the mass spectrum (or its concentration was below the detection limit). The measured intensities for $BrO^-$ and $BrO_2^-$ were 0.003 and 0.001 ions $s^{-1}$, respectively, while the $Br^-$ and $H_2OBr^-$ intensities were 129 and 32 ions $s^{-1}$, respectively. Therefore, the sum of Br- and $H_2OBr^-$ is 40 000 times larger than the $BrO^-$ and $BrO_2^-$; the influence of $BrO^-$ and $BrO_2^-$ on overall detection should be very small.

Additionally, if we assume $BrO^-$ and $BrO_2^-$ have the same sensitivity as the $I_2$, the total concentration of $BrO^-$ and $BrO_2^-$ will be $6.7\times10^5$ molec $cm^{-3}$. With a residence time of 20 ms in the ion-molecule reaction chamber of the MION

inlet and an ion-molecule collision rate of $2\times10^{-9}$ cm$^3$ molec$^{-1}$ s$^{-1}$, the first order production rate of the resultant charged clusters (analyte molecule charged by BrO$^-$ or BrO$_2^-$) is around $2.7\times10^{-5}$ s$^{-1}$. Therefore, the analyte would need to be at ppbv levels to yield detectable signals in the Br-MION-CIMS, which is orders of magnitude higher than the oxidized iodine species produced in our experiments. Therefore, we conclude that the BrO$^-$ and BrO$_2^-$ (O$_3$) have a minor effect on the overall detection.

*In figure 4 and elsewhere the sensitivities are given as a concentration over signal. This figure should be flipped as the signal in the instrument is dependent on the concentration not the other way around. It would be helpful if values for sensitivity were given in the convention standard for atmospheric mass spectrometry as signal/concentration.*
Reply: We have swapped the axis according to the reviewer's suggestion.

*I can summarize my main concern with respect to the authors not presenting the errors associated with the work appropriately with one example. There is a large discrepancy between the permeation source derived I2 sensitivity and the DOAS derived sensitivity 2.7e10 versus 6.3e10. That is a factor of 2 difference! It seems this is just glossed over and needs to be explained. These are even measured sensitivities with that large of a discrepancy so I am inclined to believe the errors in the approximations are at least as large and likely significantly larger than this difference.*
Reply: The reviewer is correct that we describe a difference of a factor of approximately 2.3 for the calibrations of I$_2$ by CE-DOAS and permeation tube calibrations. However, these two calibrations were carried out with different instruments and slightly different settings, at different times. Due to limitations of available equipment, we were unable to compare the two methods simultaneously. The manuscript was insufficiently clear in this regard. The main take-home message is that in-situ calibration is critical. In this context we regard the agreement within a factor of 2.3 as reasonable; these tuning differences require in-situ calibration but indicate that the instruments are consistent, but the final factor of 2 requires careful calibration. In our case, one of the calibrations (by CE-DOAS) was done onsite at CLOUD while the off-site calibration (by a permeation tube) was carried out in a laboratory at the University of Helsinki. The reviewer is also right that it is misleading to state that the results are consistent. Our initial intention was stating that calibration factors were not inconsistent, since a factor of two difference in two independent calibrations is common in the characterization of the chemical ionization method we use. Three changes are made in the revised version:

1. We have deleted the statement about the consistency in the calibration coefficients;
2. We added discussions about the difference in the inter-method (in-situ) and offline calibrations and highlighted the importance of carrying out calibration experiments at individual experiments;
3. We added a table which lists the key parameters of the Br-MION-CIMS in the inter-method and offline experiments.

*Specific comments:*
*- Section 3.3.4 and 3.3.5 should be flipped as the first sentence of 3.3.4 states the calibration was performed as in 3.3.5*

Reply: We have reconstructed the section 3.3, and now the $H_2SO_4$ calibration appears before HOI calibration.

*- Figure 2 caption doesn't seem to have any proton transfer reaction products as the caption would lead the reader to believe.*

Reply: We only show selected iodine species in Figure 2, as we state in the caption. We therefor deem it sufficient to leave things as is.

*- Figure 6, as sulfuric acid is one of the main foci of this manuscript it really should be included in this figure to show the response time of the measurement.*

Reply: The reviewer makes a good point. However, we do not have $H_2SO_4$ measurements from the Br-FIGAERO-CIMS – the FIGAERO-CIMS was switched to iodide chemical ionization scheme at that time. We now have removed the Fig 6 and condensed the discussion as per reviewer 1's request.

**Table S1: Instrument specifications for Br-MION-CIMS and Br-FIGAERO-CIMS.** The values are specific to our instruments, thus can vary according to instrument parameters.

| Instrument specifications | Br-MION-CIMS (lab) | Br-MION-CIMS (CLOUD) | Br-FIGAERO-CIMS |
|---|---|---|---|
| Total sample flow (slpm) | 20 | 32 | 18 |
| IMR pressure (mbar) | 1000 | 1000 | 150 |
| IMR residence time (ms) | 30 | 20 | 200 |
| SSQ pressure (mbar) | 1.9 | 2.2 | 2.0 |
| BSQ pressure (mbar) | 0.011 | 0.012 | 0.011 |
| [a]MION accelerator (V) | -2750 | -2800 | n/a |
| [b]MION deflector (V) | -210 | -290 | n/a |
| Nozzle (V) | 11.4 | -1.4 | 0.05 |
| SSQ EP (V) | 13.79 | -5.13 | 0.10 |
| SSQ front (V) | 20.92 | 12.45 | 0.20 |
| SSQ back (V) | -16.94 | 8.22 | -0.30 |
| Lense skimmer (V) | -16.17 | -6.16 | -0.19 |
| Skimmer (V) | 0.59 | 0.63 | 4.00 |
| BSQ front (V) | 4.78 | 1.76 | 5.89 |
| BSQ back (V) | 4.93 | 2.76 | 5.90 |
| Skimmer 2 (V) | 5.77 | 5.69 | 5.92 |
| Reference (bias) (V) | 124.94 | 105.87 | 121.99 |
| Ion-lense (V) | 34.076 | 31.21 | 39.78 |
| Deflector flange (V) | 53.79 | 75.91 | 88.31 |
| Deflector (V) | 61.77 | 83.82 | 95.08 |
| Mass analyzer | microchannel plate detector | microchannel plate detector | microchannel plate detector |
| Mass resolution | ~ 10000 | ~ 10000 | ~ 10000 |

[a] The voltage to accelerate the reagent ions in the ion source (Rissanen et al. 2019).

[b] The voltage to deflect the reagent ions to the sample flow (Rissanen et al. 2019).

**References**

Passananti, M., Zapadinsky, E., Zanca, T., Kangasluoma, J., Myllys, N., Rissanen, M. P., Kurtén, T., Ehn, M., Attoui, M. and Vehkamäki, H.: How well can we predict cluster fragmentation inside a mass spectrometer?, Chem. Commun., doi:10.1039/c9cc02896j, 2019.

Tham, Y. J., He, X.-C., Li, Q., Cuevas, C. A., Shen, J., Kalliokoski, J., Yan, C., Iyer, S., Lehmusjärvi, T., Jang, S. and others: Direct field evidence of autocatalytic iodine release from atmospheric aerosol, Proc. Natl. Acad. Sci., 118(4), 2021.